# Emissions mitigation opportunities for savanna countries from early dry season fire management

Geoffrey J. Lipsett-Moore [1], Nicholas H. Wolff [2] & Edward T. Game[1,3]

Savanna fires produce significant emissions globally, but if managed effectively could provide an important mitigation opportunity, particularly in African least developed countries. Here we show global opportunities for emissions reductions through early dry season burning for 37 countries including: 29 countries in Africa (69.1 MtCO$_2$-e yr$^{-1}$), six countries in South America (13.3 MtCO$_2$-e yr$^{-1}$), and Australia and Papua New Guinea (6.9 MtCO$_2$-e yr$^{-1}$). Emissions reduction estimates are based on the successful approach developed in Australia to reduce emissions from savanna fires using global-scale, remotely sensed estimates of monthly emissions. Importantly, 20 least developed countries in Africa account for 74% of the mitigation potential (60.2 MtCO$_2$-e yr$^{-1}$). More than 1.02 million km$^2$ of savanna dominated protected areas within these countries could be used as pilot sites to test and advance a regional approach to mitigation efforts for savanna fires in Africa. Potential versus actual abatement opportunities are discussed.

[1] The Nature Conservancy, Asia Pacific Resource Centre, 48 Montague Road, Level 1, South Brisbane, QLD 4101, Australia. [2] The Nature Conservancy, Global Science, 14 Maine Street, Suite 401, Brunswick, ME 04011, USA. [3] University of Queensland, Brisbane, QLD 4072, Australia. Correspondence and requests for materials should be addressed to G.J.L-M. (email: geoff.lipsett-moore@tnc.org)

The savanna biome contributes 30% of terrestrial net primary production[1], spans 20% of the Earth's terrestrial surface[2], and is home to 20% of the human population and the majority of livestock[3]. Savannas are also the world's most fire prone landscapes. Savanna fires contributed 62% (4.92 PgCO$_2$-e yr$^{-1}$) of gross global mean fire emissions, where total global fire emissions were 8.07 PgCO$_2$-e yr$^{-1}$ between 1997 and 2016[4]. Although regrowth from vegetation postfire tends to sequester the carbon dioxide (CO$_2$) released into the atmosphere[5], methane (CH$_4$), and nitrous oxide (N$_2$O) emissions persist in the atmosphere and contributed an approximate net of 2.1 PgCO$_2$-e yr$^{-1}$, or the equivalent of 6% of global fossil fuel emissions in 2014[6]. Reducing savanna fire emissions represents a potentially important mitigation opportunity toward reaching the Paris Agreement goal of maintaining global temperatures below 2 °C of warming this century[7]. While management of fire season has been proposed as a general climate mitigation instrument[8, 9], the scope and scale of the opportunity is unclear.

The first project aimed at reducing green-house gas (GHG) emissions from savanna burning was established in northern Australia, the 24,000 km$^2$ Western Arnhem Land Fire Abatement project (WALFA). The approach was to intentionally burn savannas during the early dry season (EDS), when fires are generally smaller, less intense, and release fewer emissions, with the goal of reducing fire occurrence, intensity, extent, and emissions late in the dry season[10–13]. Following years of capacity building and emissions research[10, 14, 15], the WALFA project became fully operational in 2005[12]. The project was based on a voluntary emissions offset program with a multinational energy corporation, to abate 100,000 tCO$_2$-e yr$^{-1}$ over 17 years at $10 AUD tCO$_2$-e. The project-specific accounting methodology also received formal endorsement from the Australian Government. The 10-year baseline (1994–2005) emissions for the WALFA project were 310,024 tCO$_2$-e yr$^{-1}$. By shifting the fire regime from an average of 7.6% of area burned early and 32% of area burned late to an average of 20.9% burned early and 10.9% burned late, the fire managers achieved a mean annual emissions reduction of 37.7% (116,968 tCO$_2$-e), relative to the baseline, over the first 7 years of operations[12]. The WALFA project, and its successors, delivers improved livelihoods for Indigenous Australians in remote areas where income generation options are extremely limited[16].

In 2014, the Australian government established the Emissions Reduction Fund (ERF)[17], which offered long-term public contracts to landowners and managers for storing carbon or reducing GHG emissions. The approved savanna burning methodology provided the vehicle for savanna landowners to engage in EDS fire management and the ERF[18]. These enabling conditions mobilized a proliferation of new savanna burning projects. As of 15 January 2018, a total of 75 savanna burning projects were registered under the ERF and 52 of these projects have secured contracts with the Australian Government to abate 13.8 MtCO$_2$-e over an average of 8.5 years[19]. Across the six auctions, and all 413 abatement projects, the published average carbon price was $11.90 AUD tCO$_2$-e yr$^{-1}$. Savanna burning projects account for 7.2% (191.7 MtCO$_2$-e yr$^{-1}$) of Australia's ERF contract portfolio, and 23 Indigenous projects account for 74% of the total potential savanna burning abatement. This is expected to provide significant incomes to Indigenous landowners over the next 7–10 years.

A crucial requirement for the scaling-up of any mitigation efforts are enabling conditions. Under the Paris Agreement, advanced economies have formally agreed to mobilize $100 billion USD per year by 2020 to address the pressing adaptation needs of developing countries through the Green Climate Fund (GCF)[7, 20]. The fund focuses its attention on the most vulnerable countries: Least developed countries (LDCs), Small Island Developing States and African States with a 50:50 balance between mitigation and adaptation investments over time. The effective implementation of the Paris Agreement will depend on countries successfully achieving and enhancing the commitments outlined in their Nationally Determined Contributions (NDCs). Currently, although there is mention of managing and preventing wildfires, there is no mention of EDS savanna burning as a primary mitigation strategy for fire prone savannas in any NDCs or Nationally Appropriate Mitigation Actions (NAMAs)[8, 21].

The primary aims of this study were to explore whether the savanna burning methodology developed and implemented successfully in Australia to reduce emissions from savanna wildfires might also apply to other countries around the world, and to determine the extent to which savanna burning could substantively contribute toward meeting the NDCs of LDCs, and other developing countries. We explored the opportunity for reducing savanna fire emissions using global, remote-sensing estimates of area burned and default emission factors that vary in step with the progress of the burning season. We defined EDS and late dry season (LDS) burning seasons using global precipitation data. Opportunity for emissions reductions was then defined as the emissions component produced in the LDS that was available to be reduced through EDS burning. Our study suggests that 37 savanna countries would benefit from this approach, including 20 LDCs in Africa. These are the most vulnerable countries to the impacts of climate change, and therefore most in need of mitigation and adaptation initiatives to enhance their resilience.

## Results

**LDS versus EDS emissions profiles**. We assessed the emissions profiles of 50 countries across the savanna biome. Globally, net CH$_4$ and N$_2$O emissions from savanna burning averaged 228.8 MtCO$_2$-e yr$^{-1}$. Of these emissions, 227.75 MtCO$_2$-e yr$^{-1}$ (or 99.5% of total emissions), were accounted for in EDS and LDS months combined. Approximately 30% of emissions occurred in the EDS (69.2 MtCO$_2$-e yr$^{-1}$) and 70% (158.6 MtCO$_2$-e yr$^{-1}$) occurred in LDS (Table 1). Of the 50 countries analysed, 34 were high intensity burning countries (HIBCs) producing >50% of their emissions in the LDS. HIBCs produced an average of 200.2 MtCO$_2$-e yr$^{-1}$, contributing 88% of global emissions from savanna fires. The remaining 16 low intensity burning countries (LIBCs) produced an average of 27.5 MtCO$_2$-e yr$^{-1}$, contributing 12% of global emissions from savanna fires. For HIBCs 26% (51.6 MtCO$_2$-e yr$^{-1}$) of emissions were produced in the EDS and 74% (148.5 MtCO$_2$-e yr$^{-1}$) of emissions were produced in the LDS. In contrast, for LIBCs 64% (17.5 MtCO$_2$-e yr$^{-1}$) of emissions were produced in the EDS and 36% (10.0 MtCO$_2$-e yr$^{-1}$) of emissions were produced in the LDS.

HIBCs are those most capable of generating abatement by reducing LDS emissions through EDS fire management. If LDS fires could be largely eliminated through EDS fire management, then the global abatement potential of HIBCs would be around 96.8 MtCO$_2$-e yr$^{-1}$, or an abatement potential of 48% of the total emissions from savanna fires (assuming a 15-year baseline) (Table 1). Across all countries, inclusive of LIBCs, the abatement potential would be 89.4 MtCO$_2$-e yr$^{-1}$, or 39% of the total global emissions from savanna fires, provided all remaining emissions from LDS fires could be abated (Table 1).

**Minimum late dry season emissions for viable projects**. While HIBCs (>50% LDS emissions) are better candidates for the approach, countries must also generate sufficient emissions in the LDS (>50,000 tCO$_2$-e yr$^{-1}$) to warrant further exploration. Using this criterion, countries generating small emissions in the LDS

**Table 1 Total emissions for late (>50% emissions LDS) and early (<50% emissions LDS) burning savanna countries (for all 50 countries >600 mm rainfall per year)**

| Late dry season vs. early dry season emissions | Countries (N) | LDCs | EDS tCO$_2$-e | LDS tCO$_2$-e | Total tCO$_2$-e | Abatement potential (AP = LDS−EDS) tCO$_2$-e |
|---|---|---|---|---|---|---|
| High intensity burning countries (HIBCs) >50% emissions LDS | 34 | 16 | 51,662,640 | 148,516,934 | 200,179,574 | 96,854,295 (48%) |
| Emissions (%) | | | 26 | 74 | 88 | |
| Low intensity burning countries (LIBCs) <50% emissions LDS | 16 | 9 | 17,523,663 | 10,047,439 | 27,571,102 | −7,476,224 |
| Emissions (%) | | | 64 | 36 | 12 | |
| Total | 50 | 25 | 69,186,302 | 158,564,373 | 227,750,676 | 89,378,071 (39%) |
| Emissions (%) | | | 30 | 70 | | |

The emissions are based specifically on the contribution of CH$_4$ and N$_2$O expressed as tCO$_2$-e yr$^{-1}$. Abatement potential is expressed as (% reduction) relative to the total emissions for a 15-year baseline. All early and late dry season emissions and SD are available in Supplementary Data 1

**Table 2 Total emissions in the early dry season (EDS) and late dry season (LDS) and abatement potential for the 37 savanna countries (>600 mm rainfall per year) generating greater than 50,000 tCO$_2$-e yr$^{-1}$ in the LDS, by region**

| Region | Countries (N) | EDS tCO$_2$-e | LDS tCO$_2$-e | Total tCO$_2$-e | Emissions abatement potential tCO$_2$-e (%) |
|---|---|---|---|---|---|
| Africa | 29 | 62,519,251 | 131,594,019 | 194,113,270 | 69,074,763 (36) |
| Emissions % | | 32 | 68 | 85 | 77 |
| African LDCs | 20 | 55,670,532 | 121,895,668 | 177,566,201 | 60,193,357 (36) |
| Emissions % | | 31 | 69 | 78 | 74 |
| South America | 6 | 2,228,870 | 15,565,301 | 17,794,171 | 13,336,431 (75) |
| Emissions % | | 13 | 87 | 8 | 15 |
| Australia/PNG | 2 | 4,320,411 | 11,261,074 | 15,581,485 | 6,940,665 (43) |
| Emissions % | | 28 | 72 | 7 | 8 |
| Total | 37 | 69,068,533 | 158,420,394 | 227,488,926 | 89,351,859 (39) |
| | | 30 | 70 | 100 | 100 |

African LDCs are also identified specifically. Emissions abatement potential = LDS − EDS expressed as tCO$_2$-e and (%) (the component of emissions from LDS fires that could be reduced)

were excluded from further analysis. This included six HIBCs and seven LIBCs. This resulted in 28 HIBCs and nine LIBCs (37 countries) with significant LDS emissions (>50,000 tCO$_2$-e yr$^{-1}$) to be considered for the final analysis. Globally, these 37 countries accounted for 99.9% (227.5 MtCO$_2$-e yr$^{-1}$) of total net emissions from all savanna fires with an abatement potential of 39% (89.3 MtCO$_2$-e yr$^{-1}$) of total net emissions (Table 2). Regionally, 77% (69 MtCO$_2$-e yr$^{-1}$) of this abatement could be achieved by 29 African countries and 74% (60.2 MtCO$_2$-e yr$^{-1}$) by 20 African LDCs (Table 2). Six South American countries have the potential to abate 15% (13.3 MtCO$_2$-e yr$^{-1}$) of global savanna emissions and Australia and PNG, together, the remaining 8% (6.9 MtCO$_2$-e yr$^{-1}$) of emissions (Table 2). Australia, where the method was developed, is currently abating around 1.38 MtCO$_2$-e yr$^{-1}$. The distribution of the global abatement potential is illustrated in Fig. 1 and is expressed as: (A) LDS–EDS emissions of the combined N$_2$O and CH$_4$ components in tCO$_2$-e yr$^{-1}$, and (B) the SD of LDS–EDS emissions. The greatest abatement potential is identified in yellow, spanning large areas of sub-Saharan Africa, South America, and Australia.

**Low intensity burning countries with significant late dry season emissions**. It should be noted that some LIBCs generate sufficient LDS emissions to warrant further assessment. For example, the Sudan generated 8.85 MtCO$_2$-e yr$^{-1}$, 65% of its emissions, in the EDS and 4.75 MtCO$_2$-e yr$^{-1}$, 35% of its emissions, in the LDS. With effective EDS fire management, these LDS emissions could still be further reduced.

**Areas suitable for piloting the approach**. Our final criterion was to identify countries with sufficient suitable area (relatively unpopulated and suitable for EDS burning) for near-term development of EDS savanna burning projects. Across the countries considered here, 39 contain 377 protected areas >1000 km$^2$ (Table 3). Ultimately, there are 35 countries with >50,000 tCO$_2$-e yr$^{-1}$ emissions in the LDS and suitable areas >1000 km$^2$ (Fig. 2). Out of the 35 countries, 25 are LDCs, and 17 of these LDCs, which all occur in Africa, have the potential for piloting an emissions reductions savanna burning approach (Table 4). Within these 17 LDCs as many as 197 candidate protected areas, covering more than 1 million km$^2$, are suitable for piloting a regional approach for EDS savanna burning (see Table 4 and Fig. 2). Collectively, these 17 LDCs have the potential to abate 37% (64.2 MtCO$_2$-e yr$^{-1}$) of global savanna burning emissions. Some other countries, for example, Sierra Leone, which has significant LDS emissions (526,922 tCO$_2$-e yr$^{-1}$), lack large protected areas to pilot the approach.

## Discussion

Savannas are the world's most fire prone landscapes. Our study demonstrates that much of the fire that does occur in savannas are LDS fires. However, this skew toward LDS fires across many savanna countries also provides an important opportunity to reduce global GHG emissions. In Australia, key enabling conditions have provided the vehicle for the development and proliferation of EDS savanna burning projects. These include: (1) a significant investment in the underpinning science[10, 14, 15], (2) the development of proof of concept pilot projects (WALFA)[12], (3)

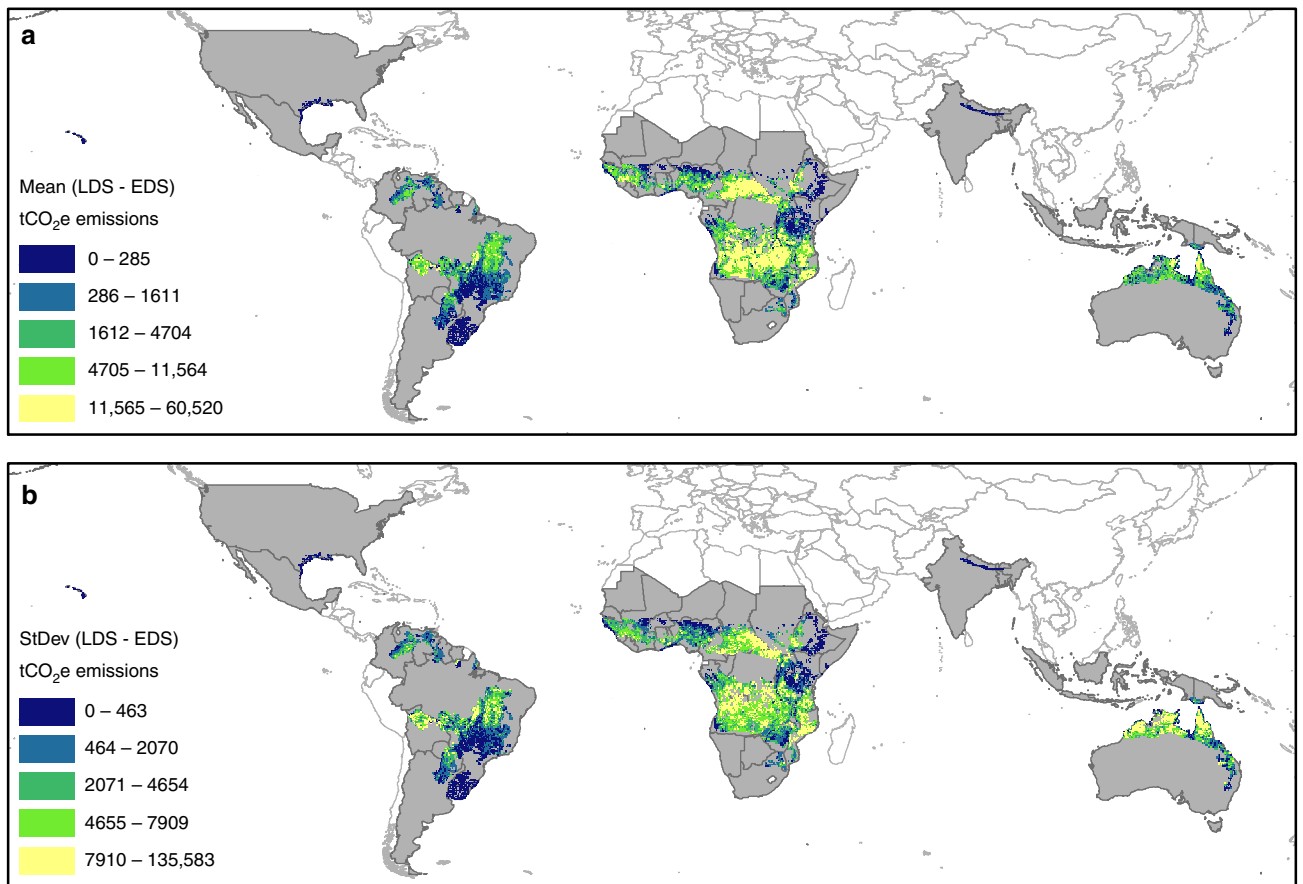

**Fig. 1** Mean annual emissions abatement potential and standard deviation. (**a**) Mean annual emissions abatement potential per pixel for the 50 countries (shaded in gray) with savanna habitat with > 600 mm rainfall yr-1, included in this study. Abatement potential is expressed as late dry season–early dry season (LDS–EDS) emissions of the combined $N_2O$ and $CH_4$ components of savanna burning, represented in $tCO_2.e$. Data categories are illustrated using quantile symbology. (**b**) Standard deviation of annual emissions abatement potential per pixel

| Table 3 Savanna countries with >600 mm rainfall per year and protected areas >1000 km² | | | | | | | | |
|---|---|---|---|---|---|---|---|---|
| Region | Countries (N) | Area of Countries (kms²) | Area of savanna (>600 mm yr⁻¹) (km²) | % | Countries (N) | Protected areas >1000km² (N) | Area of protected >600 mm yr⁻¹ >1000 km² (km²) | % |
| Africa | 35 | 20,913,585 | 8,800,336 | 56 | 28 | 258 | 1,241,033 | 68 |
| African LDCs | 25 | 16,166,159 | 7,017,103 | 45 | 17 | 203 | 1,083,978 | 59 |
| North America | 2 | 11,422,670 | 80,515 | 1 | 1 | 1 | 4580 | 0 |
| South America | 8 | 15,173,189 | 5,282,874 | 34 | 7 | 75 | 308,063 | 17 |
| Australia/SE Asia | 5 | 13,327,053 | 1,469,901 | 9 | 3 | 43 | 280,687 | 15 |
| Total | 50 | 60,836,497 | 15,633,626 | 100 | 39 | 377 | 1,834,363 | 100 |

All savanna areas by country, and protected areas by country, are available in Supplementary Data 2 and 3 respectively.

the development of a robust regulatory carbon markets such as the Carbon Farming Initiative (CFI) and ERF[17, 22], (4) a requisite carbon price (>$10–24.50 $tCO_2$-e), (5) the development of an effective EDS savanna burning emissions abatement methodology[18, 23], (6) extensive and capable Indigenous ranger programs[24], and (7) indigenous landholders and pastoralists actively seeking to generate and diversify incomes through carbon benefits[25]. By 2017, this has resulted in 54 contracted projects that will deliver 1.38 $MtCO_2$-e $yr^{-1}$ across almost 40 million hectares of the savannas of northern Australia. Comparable abatement opportunity exists for the countries identified in this analysis, including the 17 African LDCs.

If all LDS fires from savannas were removed from the 37 countries identified in this study, then EDS burning could reduce emissions by an estimated 39% (89.3 $MtCO_2$-e $yr^{-1}$) on a 15-year

pre-project baseline. Similarly, 17 African LDCs could reduce emissions by 37% (64.2 $MtCO_2$-e $yr^{-1}$) on a 15-year pre-project baseline. This is comparable to the WALFA project which has delivered an average abatement of 37.7% over 7 years, based on a 10-year pre-project baseline[12] for Australian savannas. If we assume the LDCs identified in this study were eligible for the $5 USD $tCO_2$-e results based payments offered by the GCF[26], and could effectively deliver EDS savanna burning to remove all LDS wildfires, then this could result in an estimated $321 million USD per year contribution to landowners in LDCs.

While significant mitigation potential exists to reduce emissions through EDS savanna burning projects, actual mitigation will depend on the interplay and alignment of key enabling conditions, specifically: (1) whether savanna burning is included as a priority mitigation and adaptation strategy within a country's

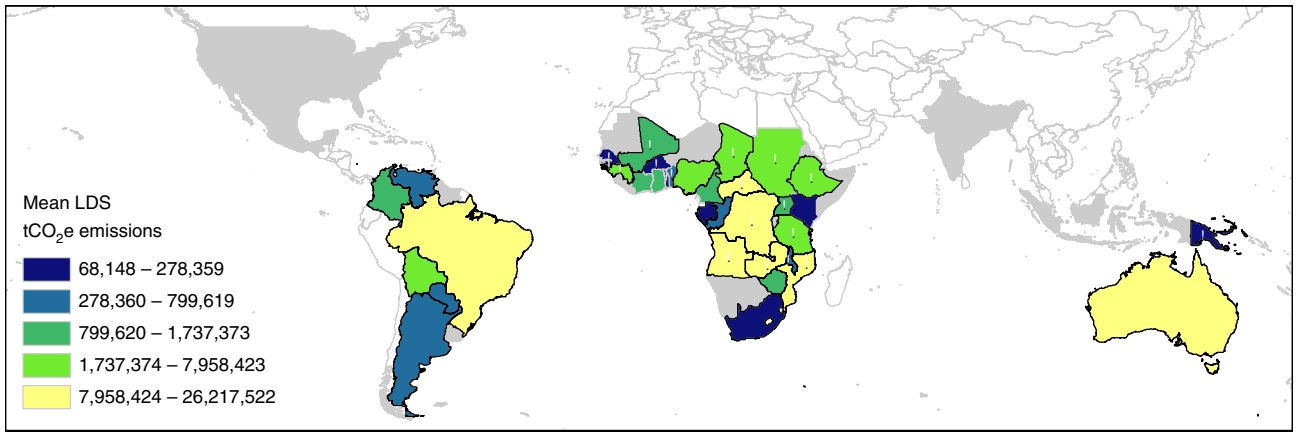

**Fig. 2** Prospective countries for early dry season savanna burning. Prospective countries for early dry season (EDS) savanna burning where colors are scaled by mean country level late dry season (LDS) emissions (using quantile bins). Heavy black outline shows countries with >50% LDS emissions and black circle symbols shows least developed countries (LDC) and small island developing (SID) countries

**Table 4** African least developed countries (LDCs) suitable for piloting EDS savanna burning where LDS emissions >50,000 $tCO_2$-e $yr^{-1}$ and Protected Areas >1000 $km^2$

| Country | Mean EDS ($tCO_2$-e $yr^{-1}$) | % | Mean LDS ($tCO_2$-e $yr^{-1}$) | % | Mean total ($tCO_2$-e $yr^{-1}$) | Mean diff ($tCO_2$-e $yr^{-1}$) | PAs >1000 $km^2$ (N) | Total area of PAs >1000 $km^2$ |
|---|---|---|---|---|---|---|---|---|
| Congo DRC | 15,984,653 | 38 | 26,217,522 | 62 | 42,202,175 | 10,232,869 | 11 | 111,468 |
| Angola | 8,668,539 | 26 | 24,361,431 | 74 | 33,029,970 | 15,692,892 | 6 | 43,419 |
| Central African Republic | 2,283,416 | 11 | 19,391,928 | 89 | 21,675,344 | 17,108,511 | 12 | 84,014 |
| Zambia | 2,801,045 | 14 | 17,293,619 | 86 | 20,094,664 | 14,492,575 | 44 | 169,939 |
| Mozambique | 6,073,121 | 39 | 9,417,789 | 61 | 15,490,910 | 3,344,668 | 19 | 76,585 |
| Sudan | 8,854,451 | 65 | 4,756,153 | 35 | 13,610,604 | −4,098,299 | 10 | 71,632 |
| Tanzania | 1,926,648 | 19 | 7,958,423 | 81 | 9,885,071 | 6,031,774 | 38 | 262,369 |
| Guinea | 1,622,061 | 37 | 2,734,593 | 63 | 4,356,654 | 1,112,532 | 4 | 40,380 |
| Ethiopia | 1,156,666 | 34 | 2,248,009 | 66 | 3,404,675 | 1,091,343 | 21 | 88,659 |
| Uganda | 1,037,038 | 45 | 1,281,073 | 55 | 2,318,111 | 244,035 | 6 | 11,317 |
| Mali | 700,574 | 41 | 1,021,034 | 59 | 1,721,608 | 320,460 | 2 | 11,815 |
| Benin | 1,023,387 | 75 | 341,300 | 25 | 1,364,687 | −682,087 | 10 | 22,633 |
| Senegal | 738,830 | 77 | 218,938 | 23 | 957,768 | −519,892 | 4 | 13,455 |
| Togo | 476,413 | 63 | 278,359 | 37 | 754,772 | −198,053 | 1 | 1,536 |
| Malawi | 308,234 | 43 | 412,799 | 57 | 721,033 | 104,565 | 2 | 4,033 |
| Burkina Faso | 566,833 | 81 | 136,732 | 19 | 703,566 | −430,101 | 5 | 8,553 |
| Guinea-Bissau | 24,875 | 7 | 344,745 | 93 | 369,620 | 319,870 | 2 | 2,408 |
| Total | 54,246,783 | | 118,414,448 | | 172,661,231 | 64,167,661 | 197 | 1,024,212 |

All emissions data by country available in Supplementary Data 1

NDCs and relevant policies and sectoral plans, (2) whether funding support is available for readiness activities to develop the capacity and infrastructure necessary for countries to participate in savanna burning initiatives including: the science to develop regional methodologies, regional data sets, and monitoring systems for National or Regional accounting[27], (3) whether there are processes and funding to mainstream EDS burning projects including: consultation, awareness-raising, capacity building, social safeguards, governance, and business models, and (4) whether there is a regulatory carbon market, or equivalent, with sufficient carbon price to incentivise the development and uptake of projects[27].

Other factors may also influence uptake of EDS savanna burning projects, particularly the decline in savanna extent through conversion of savanna to agriculture[28]. Consequently, there may be a significant difference between the technical potential abatement and the actual achievable abatement through EDS savanna burning projects. Despite these non-trivial impediments to the implementation of savanna burning projects,

managing how, and when, savannas burn will determine biodiversity, social, cultural, carbon, and economic outcomes. Therefore, the first step is to develop proof of concept, demonstration sites using large savanna protected areas in LDCs[12, 29]. Pilot projects, with appropriate biodiversity, social and cultural monitoring, could lead to wider adoption and uptake in surrounding savanna areas as the awareness of financial, social, cultural, and ecological costs and benefits are revealed[30].

For the purpose of this study, we employed a simple seasonal dichotomy of fire management in relation to carbon benefits (EDS versus LDS fire management). Using this approach imposes an artificial distinction between EDS and LDS fire emissions. Significant variation in fire intensity can occur throughout the year depending on fuel availability and conditions. The monthly Global Fire Emissions Database, Version 4.1 (GFEDv4)[31] estimates used in this study are coarse and would benefit greatly from finer scale regional data sets and validation.

The reality of fire management is extremely complex. Fire size, season, return interval, and intensity all have a profound

influence on abatement, woody thickening (sequestration)[32], pyrodiversity and biodiversity[33], and carbon pools[32]. Many countries have committed to climate change mitigation and adaptation goals, under the Paris Agreement[7] as well as commitments to the Sustainable Development Goals (SDGs)[34]. Developing a more integrated way of including fire management will greatly assist fire prone savanna countries in meeting multiple environmental, social and economic goals.

EDS savanna burning is one option in a continuum of fire management options. Intensive fire suppression may be a key management strategy if, for example, the objective is to increase carbon sequestration through woody thickening, as per Reduced Emissions from Deforestation and Forest Degradation (REDD+) projects. If REDD+ projects are developed in fire prone savannas then this could increase risk of loss of sequestration and permanence due to LDS wild fires[35]. If maintaining open savannas for large herbivores, biodiversity conservation and tourism are key objectives then promoting LDS fire management to reduce woody thickening might be the most appropriate strategy[36]. If, however, the objectives are to reduce emissions, manage the risk of wildfire and potentially enhance biodiversity and livelihoods, then EDS fire management might be the preferred strategy. EDS fire management is not a panacea for all savanna landscapes. However, in the right circumstances EDS prescribed fire can provide a powerful emissions reduction pathway with multiple benefits[8, 16].

An ecoregions approach to savanna classification was chosen to reflect the original extent of natural savanna communities prior to major land-use change[37]. Equally valid classifications also exist. For example, approaches using current landcover also capture areas converted from forest to grasslands and thus reflect different savanna distributions than considered here[27]. Regardless of how savannas are classified and distributed, the application of the approach is still relevant.

African LDCs are among the poorest and most vulnerable countries in the world. Climate change impacts, including droughts, floods, and famines are expected to increase in these regions, exacerbating already precarious social and ecological conditions[38]. Building more climate resilient communities, livelihoods and environments, particularly in LDCs, is a major global priority. In addition to the potential mitigation benefits of savanna burning, the effective use of fire can also be an important climate change adaptation strategy for the rural poor. When fire is managed effectively, additional benefits include reduced threat of wildfires[12], maintaining and enhancing fuel wood, and nurturing and managing crops[39–41]. When fire is managed poorly, negative impacts ensue, including uncontrolled LDS wildfires, loss of crops, livestock, fuel wood, infrastructure and reduced human health and livelihoods[42]. Some fire prone LDCs and African States, for example Zambia[43], identify management of wildfires within their National Adaption Programme of Action (NAPA). However, no fire prone savanna country, or LDC, currently includes EDS savanna burning as a key climate change mitigation and/or adaptation strategy within their NDCs, NAMAs[21], or NAPAs.

Although savanna burning is an accountable activity under the provisions of the Kyoto Protocol, Australia is the only country that currently accounts for emissions from the burning of savanna in its national accounts. Currently, Australia's National GHG Inventory (NGGI) accounts for the $CH_4$ and $N_2O$ components of savanna fire emissions[10]. As our results attest, the EDS savanna burning approach provides significant abatement opportunities for many other global savanna regions. However, it is important to point out that additional emission mitigation opportunities also exist for EDS fire management, by accounting for other carbon pools. For example, Australia is actively

developing methods that account for the sequestration of the dead woody carbon pool[44]. Methods for African fire management have also been developed for sequestration from living biomass[45] and soil carbon[46]. The combination of all four carbon pools for the same fire management could significantly increase the measurable mitigation potential and revenue from EDS savanna burning projects. These combined GHG mitigation methodologies for the same EDS fire management could further enhance the viability of projects and the economic resilience of LDC communities. In addition, these same opportunities may also exist for semiarid, fire prone landscapes (150–600 mm rainfall per year) and further research and development in this area could greatly expand the geographic scope of EDS fire management opportunities identified here[47, 48].

Globally, there is a need to accelerate climate change mitigation action to keep global warming below 2 °C[49]. Equally, there is recognition of a diverse array of natural climate solutions available to assist in the process[9]. Effective EDS fire management could serve as a powerful natural climate solution to assist with the climate change mitigation and adaptation challenges facing countries within fire prone savannas. Mobilizing GCF funding to support regional pilot projects, and testing of the approach within large savanna national parks in LDCs, would provide a robust foundation for scaling-up the approach. If successful, EDS savanna burning could become one of the key initiatives providing an immediate and practical solution to reduce emissions, manage risk and improve livelihoods for some of the world's most vulnerable countries.

## Methods

**Global savannas and monthly rainfall patterns**. A global terrestrial ecoregions classification was used to delineate the global and national extent of tropical and subtropical grasslands, savannas, and shrublands[37]. The monthly mean precipitation profile of each of the 50 savanna countries included was estimated using the savanna pixels from the Global Precipitation Climatology Centre (GPCC) 0.5°, long-term (1981–2010) monthly mean precipitation dataset. Analysis was focused on just those savanna pixels with a mean annual precipitation greater than 600 mm yr$^{-1}$. Prior work in Australia has demonstrated, below 500 mm yr$^{-1}$ rainfall, fire regimes are particularly variable and fire is more likely to consistently occur when rainfall exceeds 600 mm yr$^{-1}$ [50]. For those countries that straddled the equator, monthly precipitation profiles were calculated separately for the portions of savanna north and south of the equator. Using the rainfall profiles of the 50 countries with savanna areas receiving greater than 600 mm rainfall per year, we defined EDS as the 5 months prior to the driest month of the year and the LDS burning as the 5 months after the driest month (inclusive of the driest month).

**Baselines and mean monthly fire emissions using GFEDv4**. Mean monthly fire emissions were estimated for a 15-year period (2000–2014) using version 4.1 of the Global Fire Emissions Database[31]. This period is broadly equivalent to the pre-project baseline period required for Australian savanna burning projects under the approved savanna burning methodology[18, 50]. Satellite-based estimates of burned area is the input that drives most of the spatial and temporal emissions patterns observed in GFEDv4. However, GFEDv4 incorporates the Carnegie–Ames–Stanford Approach (CASA) biogeochemical model that estimates fuel loads and combustion completeness for each monthly time step, which also influences spatiotemporal emissions patterns. The most recent version of GFEDv4, as used here, was particularly well suited for our analysis. Improvements in this version include higher spatial resolution (0.25°), updated burned area estimates that include small fires, revised fuel consumption parameterization based on field observations and improved fuel consumption estimates in frequently burning landscapes[4]. Although the inclusion of small fire data is an important improvement over prior GFED releases, it is recognized substantial uncertainties remain over the quantification of small fire emissions. Small fire uncertainty is due, in part, to the spatial resolution limitations of Moderate Resolution Imaging Spectroradiometer (MODIS) data used for quantification. While burned area estimates derived from high-resolution imagery (e.g., Landsat) will improve small fire emission uncertainty in the future, a public, global-scale database is not yet available[4].

A 15-year time series was used to correspond with the low rainfall (600–1000 mm) baseline established in Australia[18]. We extracted the mean monthly component of dry matter savanna fire emissions for each GFEDv4 pixel that intersected the savanna biome. We then used GFEDv4 emissions factors (grams of GHG species emissions per kg dry matter burned) to calculate accountable

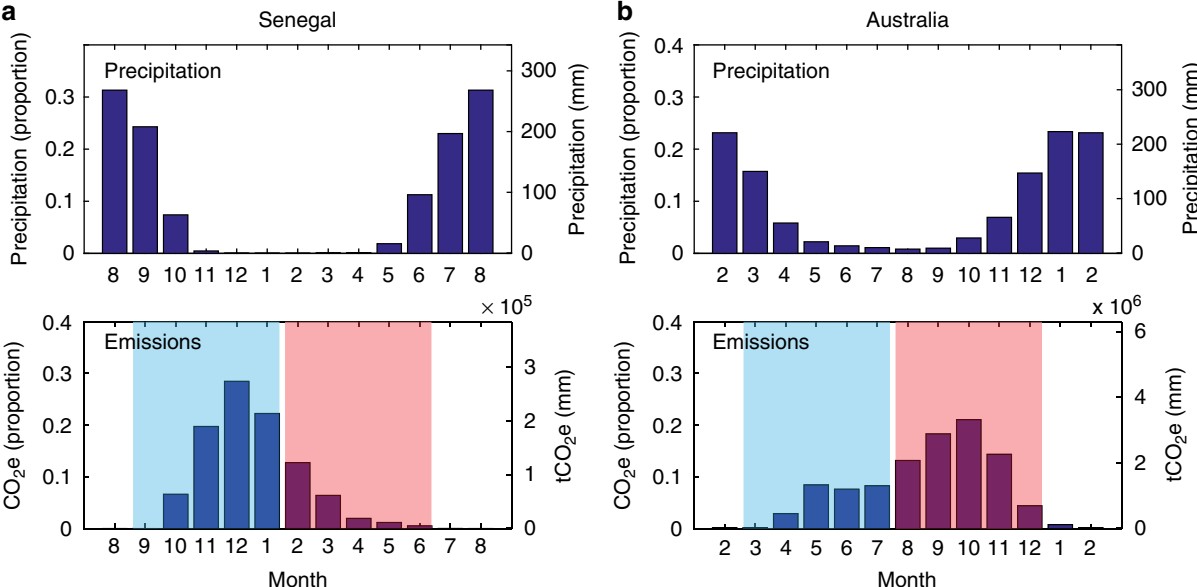

**Fig. 3** Representative example of early dry season versus late dry season patterns of savanna burning. Senegal (**a**) is an example of a low intensity burning country, where the bulk of emissions are produced from low intensity fires prior to the driest month. Australia (**b**) is an example of a high intensity burning country where the bulk of emissions are produced from high intensity fires during, or following, the driest month. For this study, early dry season (EDS) emissions are calculated using the 5 months prior to the driest month (shaded in blue) and late dry season (LDS) emissions are calculated using the 5 months following the driest month, inclusive of the driest month (shaded in red). The driest month was defined using long term (1981–2010) monthly mean precipitation. For display purposes, precipitation data are centered on the driest month for both Senegal (March) (**a**) and Australia (August) (**b**). Monthly mean emissions (2000–2014) of the combined $N_2O$ and $CH_4$ components of savanna burning are represented in $tCO_2$-e and are also expressed as a proportion of annual emissions

emissions of $CH_4$ and $N_2O$ emissions[51] per pixel. Finally, we calculated the total (sum of all corresponding pixels) monthly emissions per country for the above species. Again, for countries that straddled the equator, we treated each section (North and South) separately. We focused our analysis specifically on $CH_4$ and $N_2O$, the two internationally accountable GHGs[51] used in the Australian method and those that are not reabsorbed in the following growing season through photosynthesis[10, 18]. We converted kilograms of $CH_4$ and $N_2O$ to $CO_2$-e using standard warming potentials[51] (25×) and (298×) emissions factors, respectively.

**Defining early dry season and late dry season burning**. The distinction between EDS burning or LIBCs and LDS or HIBCs is illustrated in Fig. 3. Those countries with low intensity burn patterns produce most of their emissions prior to the driest month (e.g., Fig 3a, Senegal) and those with a high intensity burn pattern produce most of their emissions during, or after, the driest month (e.g., Fig 3b, Australia).

EDS fire management is defined as fire management carried out with the specific objective of abating emissions from fire. EDS fire management typically involves the application of a strategic EDS fire regime to reduce the risk of the occurrence, and extent of, LDS fires, through the planning and implementing of burning practices that reduce fuel loads[18]. Consequently, there is a reduction in emissions of $CH_4$ and $N_2O$ produced from fires. Under the approved savanna burning method, project abatement potential is ultimately dependent on the magnitude of baseline emissions (the total mean annual emissions from both EDS and LDS over a 15-year baseline) and the efficacy of prescribed EDS fire management in reducing extensive LDS wildfires, and their associated emissions.

**Ranking country suitability for the approach**. We ranked countries by potential suitability for the application of the approved Australian Savanna Burning method[18] using the following criteria:

First, we used the difference between the relative contribution of EDS and LDS emissions as a simple indicator of suitability. Those countries with >50% emissions in the LDS are generally more suitable for the application of the approach than those with <50% emissions in the LDS.

Second, countries need to generate a minimum of >50,000 $tCO_2$-e $yr^{-1}$ in the LDS before they would be considered suitable for project development. Viable abatement opportunity requires sufficient LDS emissions. We defined a viable project as a project capable of abating >50,000 $tCO_2$-e $yr^{-1}$ at $5 USD $tCO_2$-e using the GCF, Pilot Program for Reduced Emissions from Deforestation and Degradation or Results-based Payments[26]. It is important to note that the 50,000 $tCO_2$-e $yr^{-1}$ LDS criteria excludes HIBCs with small LDS emissions, but includes

LIBCs countries with large LDS emissions. Abatement for LIBCs would require very targeted EDS burning to further reduce LDS fires.

Finally, a crucial consideration for the development of EDS savanna burning projects is that sufficient land is available for implementation. Northern Australia is characterized by low human population density and many of the current registered and contracted projects are on Indigenous land, large conservation properties or large pastoral properties often >1000 $km^2$. Accordingly, we used the World Database on Protected Areas (WDPA) to identify protected areas >1000 $km^2$ within the global savanna biome with >600 mm $yr^{-1}$ rainfall, to define those countries with the largest near-term opportunities. The most prospective countries are those that met all our criteria: countries with >50% LDS emissions, >50,000 $tCO_2$-e $yr^{-1}$ in the LDS and with protected areas >1000 $km^2$.

**Data availability**. All data used in this article are publicly available and the data analysed and supporting the findings of this study are available from the corresponding author on reasonable request. Primary datasets used in the analyses include: GPCC 0.5°, long-term (1981–2010) monthly mean precipitation dataset (https://www.esrl.noaa.gov/psd/data/gridded/data.gpcc.html); GFED4, and WDPA. Additional Supplementary Data 1, 2 and, 3 are also available.

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

## Author contributions

G.L.M. conceived the study. G.L.M. and N.H.W. collaborated on the analysis of the data sets and N.H.W. conducted the analysis. G.L.M. and N.H.W. shared responsibility for writing the manuscript with E.T.G. All authors contributed to the final version of the manuscript.

## Additional information

**Competing interests:** The authors declare no competing interests.

