## [Peer Review File · Nature Communications]

Reviewers' comments:

Reviewer #1 (Remarks to the Author):

Thank you for the opportunity to review the paper "Global opportunities for savanna fire emissions mitigation". The paper tries to contribute to an important area in literature. However, at this stage, there are many things to improve. At this stage, I have the following suggestions that may improve the quality of the paper.

- Introduction

- o The start of the introduction can be much more effective and impressive if broader terminology like "fire as ecological factor" and "role of fire in human civilisation" are used/discussed.

- o In some places, and for some important claims, citations are missing. For example, L38-39, "Savannas support about 10% of the human population and span one-sixth of the world's land surface (@25 million km²). The savanna biome spans many of the world's Least Developed Countries (LDC's)"

- o Line 39-41, "The savanna biome spans many of the world's Least Developed Countries (LDC's). LDC's have contributed least to the emissions of greenhouse gases"; the second sentence has not backed up the first sentence and the overall theme of the paper.

- o L43-45, it is not clear whether the authors have considered all three land-based GHGs or simply N₂O and CH₄. Discussion and clarification is necessary.

- o L45-46, "Fire and its emissions can have enormous deleterious effects on human health, biodiversity and the economy". The comments about biodiversity are controversial. There are several publications that suggest that the altered fire regimes, such as long-term fire exclusion, can change vegetation patterns and lead to a biodiversity decline. Discuss following literatures:

- ♣ Andersen A. N., Hertog T. & Woinarski J. C. Z. (2006) Longterm fire exclusion and ant community structure in an Australian tropical savanna: congruence with vegetation succession, *Journal of Biogeography*, 33, 823-32.

- ♣ Bond W. J. & Parr C. L. (2010) Beyond the forest edge: ecology, diversity and conservation of the grass biome, *Biological Conservation*, 143, 2395-404.

- ♣ Maraseni, T.N., Reardon-Smith, K., Griffith, G., Apan, A. (2016). Savanna burning methodology for fire management and emissions reduction: a critical review of influencing factors, *Carbon Balance and Management*, 11:25, DOI 10.1186/s13021-016-0067-4

- ♣ Scott K., Setterfield S. A., Douglas M. M., Parr C. L., Schatz J. & Andersen A. N. (2012) Does long-term fire exclusion in an Australian tropical savanna result in a biome shift? A test using the re-introduction of fire, *Austral Ecology*, 37 (6), 693-711

- o Line 46-48, "However, in Australia, fire has once again become a tool to provide significant multiple benefits to indigenous people." This is not exclusive to Australia, with several indigenous groups and tribes across the world using fire as a tool. Some examples could be from countries which practice pastoralism and slash and burn cultivation.

- o L47-49 "Between 1990 and 2005, an average of 36% p.a. of Australia's savanna region was burnt, predominantly by late dry season (LDS) wildfires (1 Aug - 30 Dec)." Use the latest data and also provide citations.

- o Please replace "CO₂E" with "CO₂-e" elsewhere.

- o L63-65, "54 projects have secured contracts with the government under the savanna burning abatement method¹³ and are expected to abate 13.8 million tonnes CO₂E over the next 10 years at an average price of \$11.83 AUD/tCO₂E or \$163.2 million". The average price may not be applicable to these savanna burning projects since we do not know what was their actual tendered prices were. Also, these contracts were awarded in different auctions with different average prices. At minimum, authors need to acknowledge this reality in the paper.

- o L67-68: "Under the method, fire management is carried out with the intention of increasing the area burnt in the early dry season (EDS), so that less area is burnt in the late dry season (LDS)". This is not the only nor the main reason. The main reason is reducing the intensity of LDS fire. A precise discussion is necessary (see following review paper).

- ♣ Maraseni, T.N., Reardon-Smith, K., Griffith, G., Apan, A. (2016). Savanna burning methodology for fire management and emissions reduction: a critical review of influencing factors, *Carbon*

Balance and Management, 11:25, DOI 10.1186/s13021-016-0067-4

o Many highly relevant papers are missing. Please discuss them. Some key papers are given in the end.

- Methodology

o L128-132 "We defined countries as suitable and viable for the approach where: (1) $\geq 55\%$ of emissions occurred in the LDS and (2) $> 50,000$ tCO₂E/yr from CH₄ and N₂O for the 15-year baseline (see Table 2). These are the countries that would be eligible for the application of the Australia method and generating sufficient abatement to warrant project development (economically viable within the voluntary carbon market at @\$3/tCO₂E)". Couple of comments here: (1) these assumptions are not credible and need intensive discussion. In Australia, CFI/ERF follows a project based approach for proving carbon credits. This is an issue of economies of scale. In Australia, a savanna burning project might have spread over a couple hundred thousand hectares whereas, in many developing and least developed countries, getting a single project of this size is near-impossible. Therefore, the transaction cost could be much higher in these countries; (2) I am not convinced with the 15 year baseline. In low rainfall areas, the 15-year baseline period may cover only one or two fire cycles and may not provide enough data for a baseline estimation. In some parts of Africa, annual average rainfall is <600 mm (Australian threshold). Where annual rainfall averages less than 600 mm fire intervals are longer, a more extended baseline period would be required. Therefore, discuss this issue and consider an extended baseline period; (3) the price of the voluntary carbon market is going down year by year and there is no guarantee whether \$3/tCO₂e still exists over the coming decades. It will be better to link this with countries' Nationally Determined Contribution and REDD+ result based payment, rather than talking about voluntary carbon market; and (4) provide references for the claims.

o In Australian savanna burning methodology, emissions are estimated by using several variables including forest types, patchiness, fire severity class, burning efficiency, fuel class/loads, emissions factors, C content and the N to C ratio. Their values are determined on the basis of long term research in Australia. I wonder how these values were determined for different countries in this research and how reliable they are.

o L140-141 "We used the WALFA equivalent (37.7%) as our second measure of abatement potential". Rather than using single project outcomes (abatement potential) for 53 diverse countries, it is better, for a given country, to use the average of many projects within that country. If this is not possible, the average of many projects across the world would provide a better outcome. Applying this blanket approach to all 53 countries undermined the reliability and applicability of findings.

o L146-148 "The total average annual emissions from all primary greenhouse gases from savanna burning (CO₂, CH₄ and N₂O) was 3.92 Gt CO₂E". Since the beginning it is confusing whether they have covered all three land-based GHGs. I assumed they considered only 2 gases (N₂O and CH₄) but this sentence raises doubt. In the introduction and abstract sections, it needs to be clearly written which gases are covered and why?

- Results

o Line 150, "Across savannas globally, net emissions from savanna burning averaged 234.7 MtCO₂E/yr." Is this value countries' averaged or some area averaged? If area averaged then some area units would be expected.

o L180-182 "The larger the proportion of emissions in the LDS, relative to the early dry season, the greater the opportunity to reduce emissions through EDS fire management to deliver emissions abatement and carbon credits". This is quite obvious and does not add value.

o There are opportunities to make the result section more concise. Also, let the tables speak for themselves and highlight only the key points.

- Discussions

o The sense of the first couple of sentences are repeated from the introduction section.

o Here and elsewhere, please use abbreviations so that you save space. For example, greenhouse gas (GHG)

o L 210, 'savanna communities' has been used in this line and it is difficult to understand what communities it is referring to? It can refer to both people whose livelihood is based on savanna as

well savanna communities in terms of ecology.

o L231-233, repeated. Avoid it.

o L225-231 provides some conditions as to why savanna based carbon farming has huge potential in Australia. It is better to discuss the implications of not having these conditions in researched countries.

o Australia has a well-developed savanna management infrastructure as they have been undertaking prescribed burning for years. Again, authors need to discuss the implications of not having required infrastructures in researched countries.

o L254, now all "Intended Nationally Determine Contributions (INDC's)" have already become NDC. Please follow the latest literature.

o L260-262 "Indigenous communities spanning 19 million ha, where previously incomes from the land were non-existent are expected to receive of \$10 million/yr over the next ten years generated by EDS savanna burning". Need reference here.

o Even with careful fire management, there is always the possibility that a large wildfire may occur. Therefore, there is a debate about whether such a large fire may wipe out the gains in avoided emissions during the unburned period. This issues need to be discussed at length.

o If I understand correctly, the estimates are technical potential, not economic potential. There is a need for discussion of these two potentials. Key findings of this discussion should be included in the abstract/conclusion sections.

o Many highly relevant papers are missing from discussions. Please discuss them. Some of the key papers are:

- Beringer J, Hutley LB, Tapper NJ and Cernusak LA (2007) Savanna fires and their impact on net ecosystem productivity in North Australia. *Global Change Biology* 13, 990-1004.

- Bradstock RA (2008) Effects of large fires on biodiversity in south-eastern Australia: disaster or template for diversity? *International Journal of Wildland Fire* 17, 809-822.

- Bradstock, R.A., Boer, M.M., Cary, G.J., Price, O.F., Williams, R.J., Barrett, D., Cook, G., Gill, A.M., Hutley, L.B.W., Keith, H., Maier, S.W., Meyer, M., Roxburgh, S.H., Russell-Smith, J., 2012. Modelling the potential for prescribed burning to mitigate carbon emissions from wildfires in fire-prone forests of Australia. *Int. J. Wild. Fire*. doi:org/10.1071/WF11023

- Brauman A, Majeed MZ, Buatois B, Robert A, Pablo A-L, Miambi E (2015) Nitrous Oxide (N₂O) Emissions by Termites: Does the Feeding Guild Matter? *PLoS ONE* 10(12): e0144340. doi:10.1371/journal.pone.0144340

- Forbes, MS, Raison RJ, Skjemstad JO. Formation, transformation and transport of black carbon (charcoal) in terrestrial and aquatic ecosystems, *Sci. Total Environ.* 2006, 370; 190-206

- Kilinc M, Beringer J, Hutley LB, Haverd V and Tapper N (2010) A quality evaluation of flux measurements above the world's tallest angiosperm forest. *Agricultural and Forest Meteorology*

- Maraseni, T.N., Reardon-Smith, K., Griffith, G., Apan, A. (2016). Savanna burning methodology for fire management and emissions reduction: a critical review of influencing factors, *Carbon Balance and Management*, 11:25, DOI 10.1186/s13021-016-0067-4

- Murphy BP, Liedloff AC, Cook GD (2014) Does fire limit tree biomass in Australian savannas? *International Journal of Wildland Fire*. 24: 1-13.

Reviewer #2 (Remarks to the Author):

This paper is an interesting example of where science and policy intersect to create a powerful economic argument for global carbon offsets designed to mitigate climate change. The paper is based on an extension of an approach that has been developed in north Australian savannas based on a number of ad hoc empirical studies that have been combined to promote the idea of climate mitigation through altering savanna fires regimes by reducing non-CO₂ greenhouse gases. So, prima facie the case for extending this 'methodology' to savannas globally seems sound. However the logic of this approach must be much better explained and justified, either as a text box or in supplementary materials.

Most fundamentally the authors need to prove that fire intensity and emissions actually vary according to the season of fire. Such independent verification can be accomplished by using MODIS FRP. I suspect that such an analysis will show that there is considerable temporal (diurnal, seasonal and annual) variation in fire intensity across the global savannas thereby questioning the organizing principle that season is a robust proxy for non-CO2 GHG savanna emissions.

The authors need to better explain how a simple index of fire season can be used calculate greenhouse gas emissions from savannas given that the actual carbon balance of various pools in this biome remain poorly quantified – this issue is acknowledge by the authors statement that: 'Additional opportunities for the same fire management also exist for other carbon pools. Sequestration methods have been developed or are currently under development to address the dead woody carbon pool, living biomass, and soil carbon. The combination of all four carbon pools for the same fire management would significantly increase the mitigation potential of projects and equally incomes.'

The use of season to alter fire regimes at large scales for carbon management is essentially a form of geo-engineering. Serious thought must be given on how this approach can interact with existing socio-ecological systems of fire management and impacts on biodiversity and ecosystem services. There remains insufficient independent research and evaluation (i.e. conducted independently from the proponents of large-scale modification of fire regimes for 'carbon farming') to understand the costs and benefits of this approach. Such caveats must be articulated in any proposal to extend the Australian approach globally.

Reviewer #3 (Remarks to the Author):

Conceptually useful contribution, but let down by generally poor presentation which, regardless, can be readily fixed up (see attached edited doc).

As noted in edited version, ms needs to (1) describe the savanna burning methodology much more carefully, and (2) pay particular attention to more expansively describing the GFEDV4 data set, its limitations, and how those data were applied--especially given the latter provides the basis for your argument.

The ms should provide commentary concerning CDM, REDD / REDD+ opportunities, especially given savannas are considered forests under FAO and IPCC definitions

Despite the above, the paper makes a useful contribution to highlighting savanna burning opportunities and hopefully stimulates further discussions especially in African LDCountries

We thank the referees and editor for their encouraging and constructive comments, which have greatly improved the manuscript. We believe that we have taken on board the key points, modifications and suggestions. **Reviewer comments are copied in full in black, our responses are in blue and our text copied from the revised manuscript is in red.**

Reviewers' comments:

Reviewer #1 (Remarks to the Author):

Thank you for the opportunity to review the paper "Global opportunities for savanna fire emissions mitigation". The paper tries to contribute to an important area in literature. However, at this stage, there are many things to improve. At this stage, I have the following suggestions that may improve the quality of the paper.

INTRODUCTION

1. The start of the introduction can be much more effective and impressive if broader terminology like "fire as ecological factor" and "role of fire in human civilisation" are used/discussed.

We agree with the reviewer that the introduction requires far greater emphasis on the role and impact of fire, particularly in relation to our current and future human civilization and particularly vulnerable communities. To this end we have entirely revised the introduction to better focus the paper on the high level opportunities and potential beneficiaries (see revised introduction)

L23-84 - Fully revised Introduction

2. In some places, and for some important claims, citations are missing. For example, L38-39, "Savannas support about 10% of the human population and span one-sixth of the world's land surface (@25 million km²).

We have refined the sentence and incorporated the necessary citations. (see red text below)

L23-25 - The savanna biome contributes 30% of terrestrial net primary production (NPP) (Grace et al., 2006), spans 20% of the earth's terrestrial surface (Scholes & Hall, 1996) and is home to 20% of the Earth's human population and the majority of rangelands and livestock (Scholes & Archer, 1997).

3. Line 39-41, "The savanna biome spans many of the world's Least Developed Countries (LDC's). LDC's have contributed least to the emissions of greenhouse gases" (Tol et al 2004); the second sentence has not backed up the first sentence and the overall theme of the paper.

We have removed this line and reference from the text and developed a more thorough section incorporating LDC's later in the introduction around enabling conditions.

See lines 67-78

4. L43-45, it is not clear whether the authors have considered all three land-based GHGs or simply N₂O and CH₄. Discussion and clarification is necessary.

In this component of the introduction we initially wanted to define the overall gross contribution of fire to the global carbon budget and then qualify the rationale for our specific focus on the approved Australian methodology and abatement gases (CH₄, N₂O). We have now revised this to explicitly focus on the net component of CH₄ and N₂O and also changed the metric specifically to CO₂-e yr⁻¹ to avoid any confusion.

L25-29 - Savannas are also the world's most fire prone landscapes. Savanna fires contributed 62 % (4.92 Pg CO₂-e yr⁻¹) of gross global mean fire emissions, where total global fire emissions were 8.07 Pg CO₂-e yr⁻¹ between 1997–2016⁴. Although regrowth from vegetation post fire tends to sequester the CO₂ released into the atmosphere⁵, CH₄ and N₂O emissions persist in the atmosphere and an approximate net of 2.1 Pg CO₂-e yr⁻¹ or the equivalent of 6 % of global fossil fuel emissions in 2014⁶.

5. L45-46, "fire and its emissions can have enormous deleterious effects on human health, biodiversity and the economy" (Bowman et al 2009). The comments about biodiversity are controversial. There are several publications that suggest that the altered fire regimes, such as long-term fire exclusion, can change vegetation patterns and lead to a biodiversity decline. Discuss following literatures:

We have removed this statement altogether and shifted all discussion of the implications of different fire management to the discussion section. We have changed the introduction to focus specifically on better framing the role fire management might play specifically in climate change mitigation and supporting livelihoods for vulnerable communities.

Again, see revised introduction lines 23-84

1. Line 46-48, "However, in Australia, fire has once again become a tool to provide significant multiple benefits to indigenous people." This is not exclusive to Australia, with several indigenous groups and tribes across the world using fire as a tool. Some examples could be from countries which practice pastoralism and slash and burn cultivation.

We have removed this sentence to focus specifically on the use of fire specifically for mitigation and potential livelihoods for vulnerable communities.

Again, see revised introduction lines 23-84

2. L47-49 "Between 1990 and 2005, an average of 36% p.a. of Australia's savanna region was burnt, predominantly by late dry season (LDS) wildfires (1 Aug – 30 Dec)." Use the latest data and also provide citations.

We have removed lines 43-53 altogether and instead replaced the section with a specific focus on the first emissions reduction early season burning proof of concept project (the WALFA project). See below.

L34-48 - The first project aimed at reducing GHG emissions from savanna burning was established in northern Australia, the 24,000 km² Western Arnhem Land Fire Abatement project (WALFA). The basic approach is to intentionally burn savannas in the early dry season, when fires are generally smaller,

less intense, and release fewer emissions, with the goal of reducing fire occurrence, intensity, extent and emissions in late in the dry season¹⁰⁻¹³. Following years of capacity building and emissions research^{10,14,15}, the WALFA became fully operational in 2005¹². The project was based on a voluntary emissions offset program with a multinational energy corporation, to abate 100,000 tCO₂-e yr⁻¹ over 17-years at \$10 tCO₂-e. The project-specific accounting methodology also received formal endorsement from the Australian Government. The 10-year baseline (1994-2005) emissions for the WALFA project were 310,024 tCO₂-e yr⁻¹. By shifting the fire regime from an average of 7.6% burned early and 32% burned late to an average of 20.9% burned early and 10.9% burned late, the fire managers achieved mean annual emissions reduction of 37.7% (116,968 tCO₂-e) relative to the baseline over the first 7 years of operations¹². It has also been proposed that the WALFA project and its successors delivered improved livelihoods for indigenous Australians in a remote areas where income generation options were extremely limited¹⁶.

3. Please replace “CO2E” with “CO2-e” elsewhere.

We have done this throughout the document as requested.

4. L63-65, “54 projects have secured contracts with the government under the savanna burning abatement method¹³ and are expected to abate 13.8 million tonnes CO2E over the next 10 years at an average price of \$11.83 AUD/tCO2E or \$163.2 million”. The average price may not be applicable to these savanna burning projects since we do not know what was their actual tendered prices were. Also, these contracts were awarded in different auctions with different average prices. At minimum, authors need to acknowledge this reality in the paper.

We have expanded and clarified the details on both the Carbon Farming Initiative and the Emissions Reduction Fund and specifically addressed the component about different auctions highlighted. The values we have specified in L62-65 are those provide by the Australian Government as the average across all auctions. <http://www.cleanenergyregulator.gov.au/ERF/Auctions-results>

L67-78 - In 2011, the Carbon Credits (Carbon Farming Initiative) Act 2011 and associated regulations were introduced enabling the production and trading of carbon credits and the development of Carbon Farming Initiative (CFI) projects in Australia¹⁷. The CFI allowed farmers and land managers to earn carbon credits by storing carbon or reducing GHG emissions on the land. These credits were then sold to people and businesses wishing to offset their emissions. The introduction of the CFI in concert with the approved savanna burning methodologies^{18,19} provided powerful enabling conditions to incentivise the development and proliferation of savanna burning projects across northern Australia. In 2014, after a change of government, the CFI transitioned to a new regulatory arrangement, the Emissions Reduction Fund (ERF)²⁰. The ERF further expanded and consolidated enabling conditions for savanna burning projects by introducing a new method which included savanna areas with up to 600mm of rainfall annually²¹ and also providing certainty for projects through long-term contracts (7-10 years). These enabling conditions mobilized further proliferation of new savanna burning projects. **Across five auctions**, 54 project savanna burning projects have secured contracts to reduce emissions under the ERF and 26 of these are indigenous projects. These projects are contracted to abate 13.8 million tCO₂-e over the next 7-10 years at an average price of \$11.83 AUD tCO₂-e, providing an estimated \$163.2 million to indigenous landholders and pastoralists.

5. L67-68: “Under the method, fire management is carried out with the intention of increasing the area burnt in the early dry season (EDS), so that less area is burnt in the late dry season (LDS)”. This is not the only nor the main reason. The main reason is reducing the intensity of LDS fire. A

precise discussion is necessary (see following review paper).

The reviewer is correct that the both intensity and area burnt are key factors. We have adjusted the description and included the reference and additional references as follows:

L35-38 - The approach is to intentionally burn savannas in the early dry season, when fires are generally smaller, less intense, and release fewer emissions, with the goal of reducing fire occurrence, intensity, extent and emissions in late in the dry season^{10–13}

6. Many highly relevant papers are missing. Please discuss them. Some key papers are given in the end.

We have re-written the introduction to focus specifically on the global opportunity for mitigation and livelihoods, so discussion of the deeper implications of fire on biodiversity, social and cultural costs and benefits is beyond the scope of this paper.

METHODOLOGY

12. L128-132 “We defined countries as suitable and viable for the approach where: (1) $\geq 55\%$ of emissions occurred in the LDS and (2) $> 50,000$ tCO₂E/yr from CH₄ and N₂O for the 15year baseline (see Table 2). These are the countries that would be eligible for the application of the Australia method and generating sufficient abatement to warrant project development (economically viable within the voluntary carbon market at @\$3/tCO₂E)”.

Couple of comments here:

13. **R1:** these assumptions are not credible and need intensive discussion. In Australia, CFI/ERF follows a project based approach for proving carbon credits. This is an issue of economies of scale. In Australia, a savanna burning project might have spread over a couple hundred thousand hectares whereas, in many developing and least developed countries, getting a single project of this size is near-impossible. Therefore, the transaction cost could be much higher in these countries;

The reviewer is correct that the CFI/ERF follows a project based approach and that many of the projects in Australia are $> 100,000$ ha. The reviewer is also correct that it may be more challenging to find projects of sufficient size and generating sufficient carbon credits to be viable in developing countries particularly Africa. In addition, it is also true that the human population across the savanna region in Australia is low relative to both African and South American Savannas. These higher population densities will also mean higher transaction costs to secure the necessary governance and business models for developing countries. However, there is also an important exception to this rule. Large savanna protected areas provide significant opportunities for the establishment of demonstration projects. For example: Kafue NP in Zambia (2.24 million ha), Serengeti NP in Tanzania (1.47 million ha), Ruaha NP in Tanzania (2.02 million ha) to name a few. These parks are also significant in that they tend to have a long history of extensive late dry season wildfires (Archibald 2016).

We analysed the Global WDPA and established the number of protected areas $> 1000\text{km}^2$ (100,000ha) across all savanna areas with $> 600\text{mm}$ of rainfall to further substantiate the likely opportunity (see Table 3 and Table 4).

We further defined countries as suitable for the approach where > 50,000 tCO₂-e/yr was generated in the LDS from CH₄ and N₂O for the 15year baseline (see Table 2), and with protected areas >1000km² receiving > 600mm of rainfall/yr. The WALFA project in Australia consistently generated 0.045tCO₂-e/ha/yr over 7 years (Russell-Smith et al 2013). At this level of abatement, a 100,000 ha protected area would be capable of generating @4500tCO₂-e/yr from EDS fire management and assuming REDD+ payments of \$5USD tCO₂-e, could generate @\$22,500 USD/yr.

See revised methods 88-178.

L171-178 – (Area based projects) Finally, a crucial consideration for the development of EDS savanna burning projects is that sufficient land is available for implementation. Northern Australia is characterised by low human population density and many of the current registered and contracted projects are on indigenous land, large conservation properties or large pastoral properties often > 1000km². Accordingly, we used the World Database on Protected Areas (WDPA) to identify protected areas >1000km² within the global savanna biome with > 600 mm yr⁻¹ rainfall, to defined those countries with the largest immediate opportunity. Clearly, the most prospective countries are those that met all of our criteria: countries with > 50% LDS emissions, > 50,000 tCO₂-e yr⁻¹ and with protected areas > 1000km².

L163-169 - Secondly, countries need to generate a minimum of > 50,000 tCO₂-e yr⁻¹ in the LDS before they would be considered suitable for project development. That is, viable abatement opportunity requires sufficient LDS emissions. We defined a viable project as a project capable of abating > 50,000 tCO₂-e yr⁻¹ at \$5USD tCO₂-e using the Green Climate Fund - Pilot Program for REDD+ or Results-based Payments²⁸. It's important to note that the 50,000 tCO₂-e yr⁻¹ LDS criteria excludes HIBCs with small LDS emissions, but includes LIBCs countries with large LDS emissions. Abatement for LIBCs would require very targeted EDS burning to further reduce LDS fires.

14. **R1** - I am not convinced with the 15 year baseline. In low rainfall areas, the 15-year baseline period may cover only one or two fire cycles and may not provide enough data for a baseline estimation. In some parts of Africa, annual average rainfall is <600mm (Australian threshold). Where annual rainfall averages less than 600 mm fire intervals are longer, a more extended baseline period would be required. Therefore, discuss this issue and consider an extended baseline period;

The reviewer is correct, for our original manuscript and analyses, we set the lower threshold for rainfall at 500mm, which is 100mm lower than the approved savanna burning methodology. In our revised manuscript we have set the minimum threshold for rainfall at 600mm congruent with the approved savanna burning methodology (see Australian Government 2015) and reanalysed the data. Under the approved savanna burning method (Australian Government 2015), a 10 year baseline is required for areas receiving > 1000mm of rainfall and a 15 year baseline is required for all projects with rainfall from 600-1000mm. We have revised all figures and tables to include this major revision of the data.

The 600mm rainfall threshold is based on a significant piece of work that provided the necessary justification for the expansion of the approved method to areas < 1000mm rainfall (see Whitehead et al 2014, 2015).

Australian Government 2015. Carbon credits (carbon farming initiative—emissions abatement through savanna fire management) methodology determination 2015. Canberra: Australian Government; 2015. <https://www.legislation.gov.au/Details/F2015L00344>

Peter J. Whitehead P.J., Russell-Smith, J. and Yates C. (2014) Fire patterns in north Australian savannas: extending the reach of incentives for savanna fire emissions abatement
The Rangeland Journal 36(4) 371-388 <https://doi.org/10.1071/RJ13129>

Also published in:

Whitehead P.J., Russell-Smith, J and Yates C (2015) Fire patterns in north Australian savannas: extending the reach of incentives for savanna fire emissions abatement in Carbon Accounting and Savanna Fire Management Brett P. Murphy, Andrew C. Edwards, C.P. (Mick) Meyer and Jeremy Russell-Smith (eds). CSIRO Publishing, Clayton South, 2015, xiv + 351pp. ISBN 9780643108516.

We have fully revised all estimates to comply with the approved 600mm rainfall threshold throughout the manuscript.

15. **Reviewer # 1** - the price of the voluntary carbon market is going down year by year and there is no guarantee whether \$3/tCO₂e still exists over the coming decades. It will be better to link this with countries' Nationally Determined Contribution and REDD+ result based payment, rather than talking about voluntary carbon market; and provide references for the claims.

Very good point by the reviewer. We agree with the reviewer that it is far better to link fire related mitigation efforts with NDCs and REDD+ results based payments rather than talking about the voluntary carbon market. Current Green Climate Fund - Pilot Programme for REDD+ Results-based Payments (as per GCF/B.17/13), specifies Countries receiving REDD+ RBPs should reinvest the proceeds in activities in line with countries' Nationally Determined Contributions (NDCs) as established under the UNFCCC Paris Agreement, REDD+ strategies, or low-carbon development plans consistent with the objectives of the GCF. GCF will provide NDCs and REDD+ RBPs at \$5 USD/tCO₂E to Emissions Reductions (ERs).

L163-169 - Secondly, countries need to generate a minimum of > 50,000 tCO₂-e yr⁻¹ in the LDS before they would be considered suitable for project development. That is, viable abatement opportunity requires sufficient LDS emissions. We defined a viable project as a project capable of abating > 50,000 tCO₂-e yr⁻¹ at \$5USD tCO₂-e using the Green Climate Fund - Pilot Program for REDD+ or Results-based Payments²⁸. It's important to note that the 50,000 tCO₂-e yr⁻¹ LDS criteria excludes HIBCs with small LDS emissions, but includes LIBCs countries with large LDS emissions. Abatement for LIBCs would require very targeted EDS burning to further reduce LDS fires.

16. **Reviewer#1** - In Australian savanna burning methodology, emissions are estimated by using several variables including forest types, patchiness, fire severity class, burning efficiency, fuel class/loads, emissions factors, C content and the N to C ratio. Their values are determined on the basis of long term research in Australia. I wonder how these values were determined for different countries in this research and how reliable they are.

The Australian methodology is based on Tier 2 (Country specific biomass data and emissions factors). In order to make the relevant global comparisons we required the best available global data sets and models. The GFED4 data set used in this study is based on satellite (MODIS) derived burned area data (as is used in the Australian method) in combination with a complex dynamic vegetation model - Carnegie-Ames-Stanford-Approach (CASA) biogeochemical model (Randerson et al 2015). This data set provides global estimates of monthly burned area, monthly emissions and fractional contributions of different fire types. The data are at 0.25-degree latitude by 0.25-degree longitude

spatial resolution and are available from July 1997 through 2014. Emissions data are available for carbon (C), dry matter (DM), carbon dioxide (CO₂), carbon monoxide (CO), methane (CH₄), hydrogen (H₂), nitrous oxide (N₂O), nitrogen oxides (NO_x), non-methane hydrocarbons (NMHC), organic carbon (OC), black carbon (BC), particulate matter 2.5 microns (PM_{2.5}), total particulate matter (TPM), and sulfur dioxide (SO₂) among others. These data are yearly totals by region, globally, and by fire source for each region.

The Australian abatement methodology focused specifically on the methane (CH₄) and nitrous oxide (N₂O) gases. We selected these specific components from the GFED4 data sets to analyse for this study.

Randerson, J.T., G.R. van der Werf, L. Giglio, G.J. Collatz, and P.S. Kasibhatla. 2015. Global Fire Emissions Database, Version 4, (GFEDv4). ORNL DAAC, Oak Ridge, Tennessee, USA. <http://dx.doi.org/10.3334/ORNLDAAC/1293>. (this version???)

17. **Reviewer # 1** - L140-141 “We used the WALFA equivalent (37.7%) as our second measure of abatement potential”. Rather than using single project outcomes (abatement potential) for 53 diverse countries, it is better, for a given country, to use the average of many projects within that country. If this is not possible, the average of many projects across the world would provide a better outcome. Applying this blanket approach to all 53 countries undermined the reliability and applicability of findings.

We take the reviewers point and recognise the limitations of the WALFA equivalent (37.7%) as a blanket indicator of abatement potential. We have removed it as an index. Instead, we have chosen to use the index of abatement potential as the difference between LDS and EDS emissions (i.e. Abatement Potential = LDS-EDS emissions expressed as %), i.e. the remaining LDS emissions available to be reduced. However, it is extremely interesting to note that our abatement potential index across all emissions was 39% (see Table 2) and across all LDC's with PA's > 1000km² (37%) – see Table 4.

18. **R1** - L146-148 “The total average annual emissions from all primary greenhouse gases from savanna burning (CO₂, CH₄ and N₂O) was 3.92 Gt CO₂E”. Since the beginning it is confusing whether they have covered all three land-based GHGs. I assumed they considered only 2 gases (N₂O and CH₄) but this sentence raises doubt. In the introduction and abstract sections, it needs to be clearly written which gases are covered and why?

Good point by the reviewer. The primary aim of the paper was to explore the global opportunity for the application of the Australian EDS savanna burning methodology, which focuses specifically on net emissions from N₂O and CH₄. We have refined the manuscript to specifically focus on net emissions.

L114-123 - We used a 15 year time series to correspond with the low rainfall (600-1000mm) baselines established in Australia 21. We extracted mean monthly component of dry matter savanna fire emissions for each GFED pixel that intersected the savanna biome. We then used GFED emissions factors (g of GHG species emissions per kg dry matter burned) to calculate accountable emissions of CH₄ and N₂O emissions²⁷ per pixel. Finally, we calculated the total (sum of all corresponding pixels) monthly emissions per country for the above species. Again, for countries that straddled the equator, we treated each section (north and south) separately. We focused our analysis specifically on CH₄ and N₂O, the two internationally accountable GHGs²⁷ used in the Australian method and those that are not reabsorbed in the following growing season through

photosynthesis^{10,21}. We converted kgs of CH₄ and N₂O to CO₂-e using standard warming potentials²⁷ (25x) and (298x) emissions factors respectively.

RESULTS

19. Line 150, "Across savannas globally, net emissions from savanna burning averaged 234.7 MtCO₂E/yr." Is this value countries' averaged or some area averaged? If area averaged then some area units would be expected.

This is the total sum of contributions from all countries averaged over 15 years (i.e. the baseline). See Table 2.

L182-183. - We assessed the emissions profiles of 50 countries across the savanna biome. Across savannas globally, net CH₄ and N₂O emissions from savanna burning averaged 228.8 MtCO₂-e yr⁻¹.

20. L180-182 "The larger the proportion of emissions in the LDS, relative to the early dry season, the greater the opportunity to reduce emissions through EDS fire management to deliver emissions abatement and carbon credits". This is quite obvious and does not add value.

We disagree, and think it adds value and reinforces the fundamental premise of the paper.

21. There are opportunities to make the result section more concise. Also, let the tables speak for themselves and highlight only the key points.

I think we have done this see revised results section L182-256.

DISCUSSION

22. The sense of the first couple of sentences are repeated from the introduction section.

We have changed the introduction, so this is no longer the case.

23. Here and elsewhere, please use abbreviations so that you save space. For example, greenhouse gas (GHG)

We've made the necessary corrections as requested.

24. L 210, 'savanna communities' has been used in this line and it is difficult to understand what communities it is referring to? It can refer to both people whose livelihood is based on savanna as well savanna communities in terms of ecology.

We have changed this as suggest.

25. L231-233, repeated. Avoid it.

We have removed the repeated sentence "This has resulted in 54 contracted projects that will deliver 1.38 million tCO₂-e/yr across almost 40 million ha of northern Australia savannas" as suggest.

26. L225-231 provides some conditions as to why savanna based carbon farming has huge potential in Australia. It is better to discuss the implications of not having these conditions in researched countries.

Good point, we have tried to qualify in the following text to describe what would be required to build the necessary architecture to support the approach.

L 298-307 - Clearly, abatement and carbon credit opportunities exists for the 37 countries identified in this analysis. However, a strong regional commitment to the development of the necessary architecture would be required including: the development of national accounts and accounting for emissions from CH₄ and N₂O from savanna burning, adaptation of methods, significant consultation and awareness raising, the development of culturally appropriate, community based governance systems, capacity building and monitoring reporting and verification systems. While this is a significant undertaking, the phased development using large savanna protected areas to allow relatively rapid (free from the constraints imposed by the complexities of populated landscapes) implementation of EDS season savanna burning projects could provide the necessary pilot demonstration and learning sites.

27. Australia has a well-developed savanna management infrastructure as they have been undertaking prescribed burning for years. Again, authors need to discuss the implications of not having required infrastructures in researched countries.

In many of the larger protected areas in Africa, similar types of infrastructure exist (that is the rangers do some savanna burning). By focusing on the larger protected areas we have also focused on those areas where capacity would be greater.

28. L254, now all “Intended Nationally Determine Contributions (INDC’s)” have already become NDC. Please follow the latest literature.

Good point, we have updated all reference from INDCs to NDC’s as suggested.

29. L260-262 “Indigenous communities spanning 19 million ha, where previously incomes from the land were non-existent are expected to receive of \$10 million/yr over the next ten years generated by EDS savanna burning”. Need reference here.

See <http://www.cleanenergyregulator.gov.au/ERF/Auctions-results>

30. Even with careful fire management, there is always the possibility that a large wildfire may occur. Therefore, there is a debate about whether such a large fire may wipe out the gains in avoided emissions during the unburned period. This issues need to be discussed at length.

It is not possible to wipe out gains in avoided emissions as it in an annual activity. This is true for sequestration, but not true for avoided emissions. The government has imposed a discount rate and risk reversal buffer on sequestration projects associated with EDS fire management, but not on the avoided emissions component. Where this becomes an issue is where large projects with multiple groups commit to abating say 100,000tCO₂E, particularly for contractual arrangements. In these instances, if one group gets burnt out the remaining groups need to shoulder the load in terms of abatement lost and contractual commitments. What we are finding in northern Australia is that the better we manage the EDS fires the less the risk of LDS wildfires.

31. If I understand correctly, the estimates are technical potential, not economic potential. There is a need for discussion of these two potentials. Key findings of this discussion should be included in the abstract/conclusion sections.

The estimates are a technical potential of what would be possible if we could achieve the estimated levels of abatement. Also, the estimates do reflect the on-ground abatement potential achieved in Australia. The economic potential in the first instance would be what might be feasible to achieve in the large protected areas. If we were able to get approved projects across all large protected areas in the LDC's identified, (i.e. 1million km² or 100,000,000 ha then this could generate @4,500,000 tCO₂E/yr @\$5USD/tCO₂E - \$22,500,000 USD/yr.

32. Many highly relevant papers are missing from discussions. Please discuss them. Some of the key papers are:

While we agree these are important papers, we have reframed the entire paper, so these deeper levels of exploration might distract from the focus of the key themes, namely mitigation opportunity globally and enabling conditions to support the approach.

- Beringer J, Hutley LB, Tapper NJ and Cernusak LA (2007) Savanna fires and their impact on net ecosystem productivity in North Australia. *Global Change Biology* 13, 990-1004.
 - Bradstock RA (2008) Effects of large fires on biodiversity in south-eastern Australia: disaster or template for diversity? *International Journal of Wildland Fire* 17, 809-822.
 - Bradstock, R.A., Boer, M.M., Cary, G.J., Price, O.F., Williams, R.J., Barrett, D., Cook, G., Gill, A.M., Hutley, L.B.W., Keith, H., Maier, S.W., Meyer, M., Roxburgh, S.H., Russell-Smith, J., 2012. Modelling the potential for prescribed burning to mitigate carbon emissions from wildfires in fire-prone forests of Australia. *Int. J. Wild. Fire*. doi:org/10.1071/WF11023
 - Brauman A, Majeed MZ, Buatois B, Robert A, Pablo A-L, Miambi E (2015) Nitrous Oxide (N₂O) Emissions by Termites: Does the Feeding Guild Matter? *PLoS ONE* 10(12): e0144340. doi:10.1371/journal.pone.0144340
 - Forbes, MS, Raison RJ, Skjemstad JO. Formation, transformation and transport of black carbon (charcoal) in terrestrial and aquatic ecosystems, *Sci. Total Environ.* 2006, 370; 190–206
 - Kilinc M, Beringer J, Hutley LB, Haverd V and Tapper N (2010) A quality evaluation of flux measurements above the world's tallest angiosperm forest. *Agricultural and Forest Meteorology*
- Included: Maraseni, T.N., Reardon-Smith, K., Griffith, G., Apan, A. (2016). Savanna burning methodology for fire management and emissions reduction: a critical review of influencing factors, *Carbon Balance and Management*, 11:25, DOI 10.1186/s13021-016-0067-4
- Murphy BP, Liedloff AC, Cook GD (2014) Does fire limit tree biomass in Australian savannas? *International Journal of Wildland Fire*. 24: 1-13.

Reviewer #2 (Remarks to the Author):

33. This paper is an interesting example of where science and policy intersect to create a powerful economic argument for global carbon offsets designed to mitigate climate change. The paper is

based on an extension of an approach that has been developed in north Australian savannas based on a number of ad hoc empirical studies that have been combined to promote the idea of climate mitigation through altering savanna fires regimes by reducing non-CO₂ greenhouse gases. So, prima facie the case for extending this 'methodology' to savannas globally seems sound. However the logic of this approach must be much better explained and justified, either as a text box or in supplementary materials.

We have provided additional information and supplementary materials to more fully support the case.

33. Most fundamentally the authors need to prove that fire intensity and emissions actually vary according to the season of fire. Such independent verification can be accomplished by using MODIS FRP. I suspect that such an analysis will show that there is considerable temporal (diurnal, seasonal and annual) **variation in fire intensity across the global savannas thereby questioning the organizing principle that season is a robust proxy for non-CO₂ GHG savanna emissions.**

We recognize there is considerable diurnal, seasonal and annual variation in the intensity of savanna fires and that this intensity influences emissions. However, the Carnegie–Ames–Stanford Approach (CASA) biogeochemical model incorporated by GFED includes estimates of fuel loads and combustion completeness in each monthly time step. In other words, seasonal/annual variability is embedded within the emissions data. Having said that, the most important driver of emissions in savanna landscapes is burned area. Importantly, the GFED version (4.1) we use incorporates smaller fires (excluded in previous versions), improving estimates of burned area. The approach we suggest, more prescribed EDS burning, reduces LDS emissions by both reducing intensity AND extent of LDS fires. We have rewritten our methods to better describe what GFED data capture:

van der Werf GR, Randerson JT, Giglio L, van Leeuwen TT, Chen Y, Rogers BM, Mu M, Marle MJE van, Morton DC, Collatz GJ, Yokelson RJ, Kasibhatla PS (2017) Global fire emissions estimates during 1997–2016. Earth System Science Data 9:697–720

Mean monthly fire emissions were estimated for a 15-year period (2000-2014) using version 4.1 of the Global Fire Emissions Database (GFEDv4)²⁶. This period is broadly equivalent to the pre-project baseline assessment period required for Australian savanna burning projects established under lower rainfall (1000 – 600 mm mean annual rainfall)²⁵. Satellite based estimates of burned area is the input that drives most of the spatial and temporal emissions patterns observed in GFED. However, GFED incorporates the Carnegie–Ames–Stanford Approach (CASA) biogeochemical model that estimates fuel loads and combustion completeness for each monthly time step, which also influences spatio-temporal emissions patterns. The most recent version of GFED, which we use here, represents significant improvements over prior versions that made this release particularly well suited for our analysis. Improvements include higher spatial resolution (0.25°), updated burned area estimates that include small fires, revised fuel consumption parameterization based on field observations and improved fuel consumption estimates in frequently burning landscapes⁴.

34. The authors need to better explain how a simple index of fire season can be used calculate

greenhouse gas emissions from savannas given that the actual carbon balance of various pools in this biome remain poorly quantified – this issue is acknowledged by the authors' statement that:

'Additional opportunities for the same fire management also exist for other carbon pools. Sequestration methods have been developed or are currently under development to address the dead woody carbon pool, living biomass, and soil carbon. The combination of all four carbon pools for the same fire management would significantly increase the mitigation potential of projects and equally incomes.'

We agree with the reviewer that the carbon balance in the different carbon pools across savannas is extremely complex, which is why we have focused on the abatement of CH₄ and N₂O component only in the first instance. There is clearly much work to be done in this area. Our primary goal in this paper was to flag the potential opportunity, which, if supported, should result in deeper exploration of the complexity and dynamics between the different carbon pools. Our simple index in the revised version focuses specifically on LDS-EDS emissions expressed as a % of the total emissions. This gives the residual LDS emission available to be reduced as a proxy for potential abatement (assuming you can remove all LDS emissions through EDS fire management). That said, it is extremely interesting to note that our simple abatement potential index across all emissions was 39% (see Table 2) and across all LDC's with PA's > 1000km² (37%) – see Table 4. In the same ballpark as the WALFA project (37.7%)

35. The use of season to alter fire regimes at large scales for carbon management is essentially a form of geo-engineering. Serious thought must be given on how this approach can interact with existing socio-ecological systems of fire management and impacts on biodiversity and ecosystem services. There remains insufficient independent research and evaluation (i.e. conducted independently from the proponents of large-scale modification of fire regimes for 'carbon farming') to understand the costs and benefits of this approach. Such caveats must be articulated in any proposal to extend the Australian approach globally.

To some extent, fire season control is already being implemented in Africa through the implementation of suppression policies. If we could plan for the phased implementation of savanna burning projects starting with the large savanna national parks we could start to test and understand the biodiversity implications of the approach. You could build out from the parks over time, working with neighbours to build awareness, capacity and governance systems. We need to include some caveats here. Also, what are the alternative sources of income and what are their impacts on biodiversity, social and cultural systems?

Reviewer #3 (Remarks to the Author):

37. Conceptually useful contribution, but let down by generally poor presentation which, regardless, can be readily fixed up (see attached edited doc).

We have included almost all the edits provided by reviewer #3 38. As noted in edited version, ms needs to:

- a. describe the savanna burning methodology much more carefully, and

We have attempted to do this as follows:

L145-155 - EDS fire management is defined as fire management carried out with the specific objective of abating emissions from fire. EDS fire management typically involves the application

of a strategic early dry season fire regime to reduce the risk of the occurrence and extent of late dry season fires, through the planning and implementing of burning practices that reduce fuel loads²¹. The overall outcome is that fires burn at lower intensities due to a reduction in available fuel for combustion, a reduction in the area burnt at high intensities, and a reduction in the proportion of fuel consumed by fire. Consequently, there is a reduction in emissions of CH₄ and N₂O produced from fires. Under the approved savanna burning method, project abatement potential is ultimately dependent on the magnitude of baseline emissions (i.e. the total mean annual emissions from both EDS and LDS over a 15-year baseline) and the efficacy of prescribed EDS fire management in reducing extensive LDS wildfires and their associated emissions.

- b. pay particular attention to **more expansively describing the GFEDV4 data set, its limitations, and how those data were applied**—especially given the latter provides the basis for your argument.

We have attempted to expand on the details of the GFED data set.

L 101-122 - Mean monthly fire emissions were estimated for a 15-year period (2000-2014) using version 4.1 of the Global Fire Emissions Database (GFEDv4)²⁶. This period is broadly equivalent to the pre-project baseline assessment period required for Australian savanna burning projects established under lower rainfall (1000 – 600 mm mean annual rainfall)²⁵. Satellite based estimates of burned area is the input that drives most of the spatial and temporal emissions patterns observed in GFED. However, GFED incorporates the Carnegie–Ames–Stanford Approach (CASA) biogeochemical model that estimates fuel loads and combustion completeness for each monthly time step, which also influences spatio-temporal emissions patterns. The most recent version of GFED, which we use here, represents significant improvements over prior versions that made this release particularly well suited for our analysis. Improvements include higher spatial resolution (0.25o), updated burned area estimates that include small fires, revised fuel consumption parameterization based on field observations and improved fuel consumption estimates in frequently burning landscapes⁴.

We used a 15 year time series to correspond with the low rainfall (600-1000mm) baselines established in Australia²¹. We extracted mean monthly component of dry matter savanna fire emissions for each GFED pixel that intersected the savanna biome. We then used GFED emissions factors (g of GHG species emissions per kg dry matter burned) to calculate accountable emissions of CH₄ and N₂O emissions²⁷ per pixel. Finally, we calculated the total (sum of all corresponding pixels) monthly emissions per country for the above species. Again, for countries that straddled the equator, we treated each section (north and south) separately. We focused our analysis specifically on CH₄ and N₂O, the two internationally accountable GHGs²⁷ used in the Australian method and those that are not reabsorbed in the following growing season through photosynthesis^{10,21}. We converted kgs of CH₄ and N₂O to CO₂-e using standard warming potentials²⁷ (25x) and (298x) emissions factors respectively.

- c. The ms should provide commentary concerning CDM, REDD / REDD+ opportunities, especially given savannas are considered forests under FAO and IPCC definitions

We have done this in terms of the REDD+ Results-based payments program.

L 163-169 - Secondly, countries need to generate a minimum of > 50,000 tCO₂-e yr⁻¹ in the LDS before they would be considered suitable for project development. That is, viable abatement opportunity requires sufficient LDS emissions. We defined a viable project as a project capable

of abating > 50,000 tCO₂-e yr⁻¹ at \$5USD tCO₂-e using the Green Climate Fund - Pilot Program for REDD+ or Results-based Payments²⁸. It's important to note that the 50,000 tCO₂-e yr⁻¹ LDS criteria excludes HIBCs with small LDS emissions, but includes LIBCs countries with large LDS emissions. Abatement for LIBCs would require very targeted EDS burning to further reduce LDS fires.

d. Despite the above, the paper makes a useful contribution to highlighting savanna burning opportunities and hopefully stimulates further discussions especially in African LD Countries

Reviewers' comments:

Reviewer #1 (Remarks to the Author):

Thank you for the opportunity to review the revised version of the paper "Global opportunities for savanna fire emissions mitigation". The paper has been improved a lot but still some minor work is necessary.

- Introduction

- o As I suggested in my comments in the earlier version, the sentence "Across five auctions, 54 project savanna burning projects have secured contracts to reduce emissions under the ERF and 26 of these are indigenous projects. These projects are contracted to abate 13.8 million tCO₂-e₂₂ over the next 7-10 years at an average price of \$11.83 AUD tCO₂-e, providing an estimated \$163.2 million to indigenous landholders and 66 pastoralists" is not correct. Firstly, there is already a sixth auction and the average price of all auctions is AUD11.90 not AUD 11.83. Secondly, the average price may not be applicable to these savanna burning projects since we do not know what was their actual tendered prices were. The average price published is based on the price of 438 different types of projects, not only from 54 savanna projects. At minimum, authors need to acknowledge this reality in the paper.

- Discussions

- o Although, it is not discussed exclusively, I understand that the emissions reduction potential estimated in this study is "technical potential". There is a huge difference between technical potential and actual potential, as the latter depends on a range of economic, social and political factors. Therefore, only a fraction of technical potential can be realised in practice. There is need for discussion about these two potentials. Key findings of this discussion should be included in the abstract/conclusion sections.

Reviewer #2 (Remarks to the Author):

This paper makes the case for using season of burning to reduce emission of CH₄ and N₂O (expressed CO₂-e yr⁻¹) from tropical savannas. The study hinges on the estimates of emissions from the version 4.1 of the Global Fire Emissions Database (GFEDv4) and is framed around the current implementation of savanna burning projects across northern Australia savannas regulated by Australian Government Emissions Reduction Fund. The paper seems technically sound and is well written.

The authors' argument hinges on the idea that savanna fire regimes can be simply and effectively toggled between early and late dry season and that there are substantial differences in fire severity (fuel consumption) between seasons. They imply that early dry season burning most probably mimics indigenous fire practices. These assumptions are, in fact, based on sketchy, and in some cases contradictory, data in northern Australia. For example, the magnitude of the difference in fire intensity between seasons is small with wide overlap (see Oliveira et al. 2015) that is smaller than differences between savanna types across the north Australian precipitation gradient. There is evidence that some Aboriginal groups in Arnhem Land set fires in the late dry season (Bowman et al. 2004), and there is a question mark as to whether contemporary early season burning emulates traditional fire practices or effectively engages with local communities (Petty et al. 2015). Further, Oliveira et al. (2015) show that fire intensity varies among broad eucalypt savanna type due to the effect of on canopy openness and grass biomass, which are inversely related. These authors concluded that "The relative intensities of fires under these different habitat conditions have significant implications for the development of savanna burning greenhouse gas emission and related carbon balance methodologies, because the rate at which plants capture and emit carbon into the atmosphere varies between different vegetation types, and the frequency and intensity of fires." Indeed, Prior et al. (2017) have shown that grass

biomass burns more completely than litter fuels, thereby affecting carbon emissions. Finally, the longer-term impacts of 'carbon farming' on tree demography and carbon pools remain poorly understood.

Given the uncertainties of the approach in northern Australia, I urge caution in applying this approach to very different socio-ecological systems across the tropics. I acknowledge that the authors have qualified their idea and recommend further research and development. I suggest however, they need to be more specific in noting that ground assessments of above ground biomass, burn severity, fuel burn completeness, and GHG emissions are required to substantiate the idea that a simple switch from late to early dry season burning can substantially reduce greenhouse gas pollution and hence climate forcing. Most fundamentally, any scheme designed to reduce GHG pollution must consider the carbon cycle, especially carbon pools and CO₂ emissions, rather than a narrow fixation of two trace gases, as is the case with the current contribution.

Bowman, D. M. J. S., Walsh, A. and Prior, L. D. (2004), Landscape analysis of Aboriginal fire management in Central Arnhem Land, north Australia. *Journal of Biogeography*, 31: 207–223. doi:10.1046/j.0305-0270.2003.00997.x

Oliveira Sofia L. J., Maier Stefan W., Pereira José M. C., Russell-Smith Jeremy (2015) Seasonal differences in fire activity and intensity in tropical savannas of northern Australia using satellite measurements of fire radiative power. *International Journal of Wildland Fire* 24, 249-260. <https://doi.org/10.1071/WF13201>

Aaron M. Petty, Vanessa deKoninck, and Ben Orlove (2015) Cleaning, Protecting, or Abating? Making Indigenous Fire Management "Work" in Northern Australia. *Journal of Ethnobiology* 35 (1), 140-162. <https://doi.org/10.2993/0278-0771-35.1.140>

Prior, L. D., Murphy, B. P., Williamson, G. J., Cochrane, M. A., Jolly, W. M. and Bowman, D. M. J. S. (2017), Does inherent flammability of grass and litter fuels contribute to continental patterns of landscape fire activity?. *J. Biogeogr.*, 44: 1225–1238. doi:10.1111/jbi.12889

We thank the referees and editor for their encouraging and constructive comments which have greatly improved the manuscript. We believe that we have taken on board the key points, modifications and suggestions. **Reviewer comments are copied in full in black, our responses are in blue and our text copied from the revised manuscript is in red** (red sections highlighted in yellow in the manuscript).

Reviewers' comments:

Reviewer #1 (Remarks to the Author):

Thank you for the opportunity to review the revised version of the paper "Global opportunities for savanna fire emissions mitigation". The paper has been improved a lot but still some minor work is necessary.

- Introduction

As I suggested in my comments in the earlier version, the sentence "Across five auctions, 54 savanna burning projects have secured contracts to reduce emissions under the ERF and 26 of these are indigenous projects. These projects are contracted to abate 13.8 million tCO₂-e₂₂ over the next 7-10 years at an average price of \$11.83 AUD tCO₂-e, providing an estimated \$163.2 million to indigenous landholders and 66 pastoralists" is not correct. Firstly, there is already a sixth auction and the average price of all auctions is AUD11.90 not AUD 11.83. Secondly, the average price may not be applicable to these savanna burning projects since we do not know what was their actual tendered prices were. The average price published is based on the price of 438 different types of projects, not only from 54 savanna projects. At minimum, authors need to acknowledge this reality in the paper.

We have revised this section in the manuscript to specifically address reviewer #1's comments and the uncertainty regarding "actual" prices received by individual projects from the different auctions. We have removed any reference to "actual" incomes to savanna burning projects and have instead focused on the expected tCO₂e to be abated with specific reference to the "published" carbon price. We have also used the most recent ERF register of projects and contracts data (15 January 2018) to update the latest numbers.

Line 54-60 As of 15 January 2018, a total of 75 savanna burning projects were registered under the ERF and 52 of these projects have secured contracts with the Australian Government to abate 13.8 MtCO₂-e over an average of 8.5 years. Across the six auctions, and all 413 abatement projects, the published average carbon price was \$11.90AUD tCO₂-e yr-1. Savanna burning projects account for 7.2% (191.7MtCO₂-e yr-1) of Australia's ERF contract portfolio, and 23 Indigenous projects account for 74% of the total potential savanna burning abatement. This is expected to provide significant incomes to Indigenous landowners over the next 7-10 years.

- Discussion

o Although, it is not discussed exclusively, I understand that the emissions reduction potential estimated in this study is "technical potential". There is a huge difference between technical potential and actual potential, as the latter depends on a range of economic, social and political factors. Therefore, only a fraction of technical potential can be realised in practice. There is need for discussion about these two potentials. Key findings of this discussion should be included in the abstract/conclusion sections.

We agree with the reviewer # 1 that the numbers we provide represent the “technical potential” for abatement. As suggested, there are a great many factors that will determine both the uptake and efficacy of the approach or the “actual” abatement. As suggested by the reviewer, we have revised our manuscript text to acknowledge the difference between these two values.

Line 301-321. While significant mitigation potential exists to reduce emissions through EDS savanna burning projects, actual mitigation will depend on the interplay and alignment of key enabling conditions, specifically: (1) whether savanna burning is included as a priority mitigation and adaptation strategy within a country’s NDCs and relevant policies and sectoral plans, (2) whether funding support is available for readiness activities to develop the capacity and infrastructure necessary for countries to participate in savanna burning initiatives including: the science to develop regional methodologies, regional data sets and monitoring systems for National or Regional accounting³¹, (3) whether there are processes and funding to mainstream EDS burning projects including: consultation, awareness- raising, capacity building, social safeguards, governance and business models, and (4) whether there is a regulatory carbon market, or equivalent, with sufficient carbon price to incentivise the development and uptake of projects³¹.

Other factors may also influence uptake of EDS savanna burning projects, particularly the decline in savanna extent through conversion of savanna to agriculture³². Consequently, there may be a significant difference between the technical potential abatement and the abatement that can actually be realised through savanna burning projects. Despite these non-trivial impediments to the implementation of savanna burning projects, managing how, and when, they burn will determine biodiversity, social, cultural, carbon and economic outcomes. The first step is the development of proof of concept, demonstration sites using large savanna protected areas in LDCs^{12,33}. These initial, pilot projects could lead to wider adoption and uptake in surrounding savanna areas as awareness of the financial, social, cultural and ecological benefits are revealed³⁴. The potential for rapid uptake, as has been demonstrated in Australia, can occur, provided key enabling conditions are established.

Reviewer #2 (Remarks to the Author):

This paper makes the case for using season of burning to reduce emission of CH₄ and N₂O (expressed CO₂-e yr⁻¹) from tropical savannas. The study hinges on the estimates of emissions from the version 4.1 of the Global Fire Emissions Database (GFEDv4) and is framed around the current implementation of savanna burning projects across northern Australia savannas regulated by Australian Government Emissions Reduction Fund. The paper seems technically sound and is well written.

The authors’ argument hinges on: (1) the idea that savanna fire regimes can be simply and effectively toggled between early and late dry season and that there are substantial differences in fire severity (fuel consumption) between seasons.

(2) They imply that early dry season burning most probably mimics indigenous fire practices. These assumptions are, in fact, based on sketchy, and in some cases contradictory, data in northern Australia. For example, the magnitude of the difference in fire intensity between seasons is small with wide overlap (see Oliveira et al. 2015) that is smaller than differences between savanna types across the north Australian precipitation gradient. There is evidence that some Aboriginal groups in Arnhem Land set fires in the late dry season (Bowman et al. 2004), and there is a question mark as to whether contemporary early season burning emulates traditional fire practices or effectively engages with local communities (Petty et al. 2015). Further, Oliveira et al. (2015) show that fire

intensity varies among broad eucalypt savanna type due to the effect of on canopy openness and grass biomass, which are inversely related. These authors concluded that “The relative intensities of fires under these different habitat conditions have significant implications for the development of savanna burning greenhouse gas emission and related carbon balance methodologies, because the rate at which plants capture and emit carbon into the atmosphere varies between different vegetation types, and the frequency and intensity of fires.” Indeed, Prior et al. (2017) have shown that grass biomass burns more completely than litter fuels, thereby affecting carbon emissions.

Finally, (3) the longer-term impacts of ‘carbon farming’ on tree demography and carbon pools remain poorly understood. Given the uncertainties of the approach in northern Australia, I urge caution in applying this approach to very different socio-ecological systems across the tropics. I acknowledge that the authors have qualified their idea and recommend further research and development. I suggest however, they need to be more specific in noting that ground assessments of above ground biomass, burn severity, fuel burn completeness, and GHG emissions are required to substantiate the idea that a simple switch from late to early dry season burning can substantially reduce greenhouse gas pollution and hence climate forcing. Most fundamentally, any scheme designed to reduce GHG pollution must consider the carbon cycle, especially carbon pools and CO₂ emissions, rather than a narrow fixation of two trace gases, as is the case with the current contribution.

We thank the reviewer for these astute and important comments. A key aim of this paper was to determine: 1) whether emissions from savanna fires globally had a distinct periodicity (EDS vs LDS) and (2) whether the periodicity of these fires might provide opportunities for countries, particularly poorer developing nations such as LDCs and African States, to apply Australia EDS methodology to reduce emissions and generate carbon credits. The reviewer is right that the EDS vs LDS dichotomy greatly simplifies complexities of fire management and potential impacts in complex socio-ecological systems. However, to highlight potential global opportunities we kept the description and definitions simple. Our hope is that the results presented here encourage countries to further explore the role that EDS fire management could play in advancing national emissions reductions commitments and diversifying mitigation beyond sequestration which has the challenge of permanence obligations.

We also agree with the reviewer that indigenous fire management is highly diverse and nuanced, and that we are not able to adequately reflect this in the space of this manuscript. We have therefore removed any reference to EDS fire management as mimicking “indigenous” fire management to avoid both the oversimplification of indigenous fire management, and the misrepresentation of indigenous fire management being an EDS fire management pattern only.

Finally, we thank the reviewer for reinforcing the point that longer-term impacts of “carbon farming” on tree demography and carbon pools remains poorly understood. We recognise that EDS fire management may not always be suitable or practical, and that for any country or community there will likely be different fire goals for different reasons, and that these goals need to be determined by the country and community in question. That said, countries and communities in fire prone landscapes need to be informed of the options available to them and to then choose the most appropriate option to meet the multiple goals of their international and national commitments.

We have revised the manuscript discussion to address these three important issues raised by Reviewer #2:

Lines 323-330. For the purpose of this study, we employed a simple seasonal dichotomy of fire management in relation to carbon benefits (EDS vs LDS fire management). The reality of fire

management is extremely complex. Fire size, season, return interval and intensity all have a profound influence on abatement, woody thickening (sequestration) 35, pyrodiversity and biodiversity³⁶ and carbon pools³⁵. Many countries have committed to NDCs and climate change mitigation and adaptation goals, under the Paris Agreement⁷ and the Sustainable Development Goals (SDGs) ³⁷. Developing a more integrated way of including fire management will greatly assist fire prone savanna countries in meeting multiple environmental, social and economic goals.

Lines – 332-342. EDS savanna burning is one option in a continuum of fire management options. Intensive fire suppression may be a key management strategy if, for example, the objective is to increase carbon sequestration through woody thickening, as per REDD+ projects. If REDD+ projects are developed in fire prone savannas then this could increase risk of loss of sequestration and permanence due to LDS wild fires³⁸. If maintaining open savannas for large herbivores, biodiversity conservation and tourism are key objectives then promoting LDS fire management to reduce woody thickening might be the most appropriate strategy³⁹. If, however, the objectives are to reduce emissions, manage the risk of wildfire and potentially enhance biodiversity and livelihoods, then EDS fire management might be the preferred strategy. EDS fire management is not a panacea for all savanna landscapes, however, in the right circumstances EDS prescribed fire can provide a powerful emissions reduction pathway with multiple benefits^{8,16}.

Reviewer 3 comments

This version much clearer and better presented than previous. Remaining issues to address:

- Reliability of GFED database still needs qualification—how reliable are estimates of burnt area and emissions of CH₄ and N₂O

Thank you for helping us get to a clearer and better presented manuscript. We used the latest GFED4 dataset (van der Werf et al. 2017). While we acknowledge the concerns of Reviewer # 3 regarding the potential reliability of the GFED4 dataset and the limitations of global data sets relative to finer scale local and national datasets, our analysis required the use of the best available global data set to make legitimate and standardized comparisons between countries and regions. It was neither feasible nor practical to use operational level National datasets such as those available in Australia (<http://www.firenorth.org.au/nafi3/>). The GFED4 dataset is considered “the most complete and current source of data on global fire emissions” according to the United Nations University’s 2015 report, *The Global Potential of Indigenous Fire Management*. The GFED4 utilises the Carnegie–Ames–Stanford Approach (CASA) biogeochemical model which incorporates both satellite data of net primary productivity (NPP) and a mechanistic plant and soil carbon model to quantify the flow of carbon through terrestrial ecosystems. Improved burned area estimates (Giglio et al., 2013; Randerson et al., 2012) and better quantification of fuel loads (Van Leeuwen et al., 2014) represent more accurate estimates of gross and net fire emissions from CO₂, CH₄ and N₂O (and other gases) than found in GFED3 (van der Werf et al, 2017).

To acknowledge and reflect remaining concerns about GFED4 limitations, we have included this text in the Methods section about GFED:

Line 104-109. Although the inclusion of small fire data is an important improvement over prior GFED releases, it is recognized substantial uncertainties remain over the quantification of small fire emissions. Small fire uncertainty is due, in part, to the spatial resolution limitations of MODIS data used for quantification. While burned area estimates derived from high-resolution imagery (e.g.

Landsat) will improve small fire emission uncertainty in the future, a public, global-scale database is not yet available⁴.

While fire CO₂ emissions – GHG implications are conventionally ignored as indicated, emissions are still in atmosphere for substantial periods/months before being resorbed—CO₂ emissions are not neutral.

The reviewer is right, and we recognise that CO₂ emissions have a significant but unquantified residual component and that post-fire re-growth and recovery may last for decades and even more than 100 years, so both current and historical fires exert impacts on land ecosystems (Amiro et al 2006, Bond-Lamberty et al 2007). However, the focus of this study was specifically on CH₄ and N₂O to avoid any confusion and uncertainty around CO₂.

Discussion section still long-winded in parts. A particular concern is assumption that all LDS fires are bad/inappropriate. For example, woody thickening / bush encroachment issues are a major land use concern in parts of southern Africa and intense fires have a significant role to play. This needs some qualification.

We have tried to reduce the length of the discussion while also responding to the requests of the reviewers.

This is an important point. EDS burning represents one fire management option in a continuum from fire suppression to LDS burning. The most appropriate type of fire management will depend entirely on the objectives set for a given savanna landscape. Implementing EDSFM for emissions mitigation could lead to woody thickening which may be a desired (increased sequestration) or undesired (decreased grazing habitat) outcome. In contrast, if only LDSFM was implemented, then the landscape would likely move to a more open savanna landscape.

We have updated our discussion to better qualify the context of when EDSFM is appropriate:

Line 332-342. EDS savanna burning is one option in a continuum of fire management options. Intensive fire suppression may be a key management strategy if, for example, the objective is to increase carbon sequestration through woody thickening, as per REDD+ projects. If REDD+ projects are developed in fire prone savannas then this could increase risk of loss of sequestration and permanence due to LDS wild fires³⁸. If maintaining open savannas for large herbivores, biodiversity conservation and tourism are key objectives then promoting LDS fire management to reduce woody thickening might be the most appropriate strategy³⁹. If, however, the objectives are to reduce emissions, manage the risk of wildfire and potentially enhance biodiversity and livelihoods, then EDS fire management might be the preferred strategy. EDS fire management is not a panacea for all savanna landscapes, however, in the right circumstances EDS prescribed fire can provide a powerful emissions reduction pathway with multiple benefits^{8,16}.

A key missing reference is the United Nations University's (2015) International Savanna Fire Management Initiative. Institute of Advanced Studies, United Nations University, Kyoto, Japan. (http://collections.unu.edu/eserv/UNU:5605/indigenous_fire_management.pdf). For example, amongst many relevant issues that report details, your supplementary Table 1 lists only Nepal and India as having areas of savanna in Asia whereas the above doc reports extensive savannas in other

regional countries including Cambodia and Myanmar (for example). I suggest that you need to qualify your savanna database selection by noting that other classification schemes are available (and equally credible)

Thank you.. We have now included a reference to the UNU Report and acknowledged that different classification approaches are both available and credible. We chose the ecoregions approach to savanna classification, where “ecoregions are relatively large units of land containing a distinct assemblage of natural communities and species, with boundaries that approximate the original extent of natural communities prior to major land-use change.” (Olson et al. 2001). We recognize that other, equally valid, classifications exist, such as those that reflect land-use change (for example, forests converted to savannas).

Lines 344-348 – An ecoregions approach to savanna classification was chosen to reflect the original extent of natural savanna communities prior to major land-use change²². Equally valid classifications also exist. For example, approaches using current landcover also capture areas converted from forest to grasslands and thus reflect different savanna distributions than considered here³¹. Regardless of how savannas are classified and distributed, the application of the approach is still relevant.

Typos etc:

L2: note that two authors are listed as corresponding authors

L28: missing ‘comprise’ at end of line

L38: delete ‘in’

L41: AUD—and apply similar conventions throughout when dealing with USD and AUD

L63: indigenous in this context should be Indigenous or Aboriginal, and check similarly throughout

L71: define LDCs, SIDS acronyms and check throughout—note sometimes you use LDC’s

L104: lower rainfall ‘conditions’...

L178: ‘define’

L209: remove parentheses around LIBCs

References: check throughout, see refs 14,15

We have addressed all typos.

REVIEWERS' COMMENTS:

Reviewer #2 (Remarks to the Author):

The authors have made some important changes to the manuscript but I would like to see a few more caveats:

1. In the Abstract it needs to be indicated that the paper rests on global-scale remote sensing estimate of monthly emissions.
2. At line 33, a concluding sentence needs to state the approach is based on global remote sensing estimates of area burn and some generic emission factors that vary in step with the progress of the burning season (see point 6 below).
3. In the Discussion, it should be stressed that that estimates of fire emissions are crude and more validation at the regional scale.
4. The word 'intense' should be dropped in the sentence in line 283. Fires in the LDS are not all intense, they are statistically more intense but there is huge variability.
5. In the sentence in line 320 it needs to be stated that the piloting of the approach needs in protected areas demands monitoring of effects on biodiversity.
6. In the paragraph starting line 325, the authors should acknowledge the using of GFED hardwires a difference between EDS and LDS fire emissions despite uncertainties in emissions and variation in fire intensity.

We once again thank the referees and editor for their encouraging and constructive comments, which have greatly improved the manuscript. We have taken on board the key points, modifications and suggestions. **Reviewer comments are copied in full in black, our responses are in blue and our text copied from the revised manuscript is in red.**

Additional style based comments and Reviewer #2 comments/edits are included in track changes in the attached final manuscript.

Reviewer #2 (Remarks to the Author):

The authors have made some important changes to the manuscript but I would like to see a few more caveats:

1. In the Abstract it needs to be indicated that the paper rests on global-scale remote sensing estimate of monthly emissions.
 - We have done this see below
 - Line 14-16 - Emissions reduction estimates are based on the successful approach developed in Australia to reduce emissions from savanna fires using global-scale, remotely-sensed estimates of monthly emissions
2. At line 33, a concluding sentence needs to state the approach is based on global remote sensing estimates of area burn and some generic emission factors that vary in step with the progress of the burning season (see point 6 below).
 - We have included this at line 79-81, where we feel it is better placed.
 - Line 79-81 We explored the opportunity for reducing savanna fire emissions using global, remote-sensing estimates of area burned and default emission factors that vary in step with the progress of the burning season.
3. In the Discussion, it should be stressed that that estimates of fire emissions are crude and more validation at the regional scale.
 - We have incorporated the changes and refinements in the discussion as requested.
 - Line 239-241 - The monthly Global Fire Emissions Database, Version 4.1 (GFEDv4)24 estimates used in this study are coarse and would benefit greatly from finer scale regional data sets and validation
4. The word 'intense' should be dropped in the sentence in line 283. Fires in the LDS are not all intense, they are statistically more intense but there is huge variability.
 - We have dropped the word intense from line 283 (now line 193 after the results were shifted to the end of the text).
 - See line 193 in track changes
5. In the sentence in line 320 it needs to be stated that the piloting of the approach in protected areas requires monitoring of the effects on biodiversity.

- We have incorporated the need for baseline and subsequent monitoring to illustrate the costs and benefits of the approach (see below). Line number shifted from line 320-line 231-234.
 - See line 231-234 - Pilot projects, with appropriate biodiversity, social and cultural monitoring , could lead to wider adoption and uptake in surrounding savanna areas as the awareness of financial, social, cultural and ecological costs and benefits are revealed³⁴.
6. In the paragraph starting line 325, the authors should acknowledge the using of GFED hardwires a difference between EDS and LDS fire emissions despite uncertainties in emissions and variation in fire intensity.
- We have adjusted the discussion to identify that we have imposed an artificial distinction between EDS and LDS fires and we have stressed that significant variation in fire intensity can occur throughout the year depending on fuel availability and conditions (see below).
 - Now Line 237-239 - Using this approach imposes an artificial distinction between EDS and LDS fire emissions. Significant variation in fire intensity can occur throughout the year depending on fuel availability and conditions.